# Collectively stabilizing and orienting posterior migratory forces disperses cell clusters in vivo

B. Lin [1]✉, J. Luo[1] & R. Lehmann [1,2]✉

Individual cells detach from cohesive ensembles during development and can inappropriately separate in disease. Although much is known about how cells separate from epithelia, it remains unclear how cells disperse from clusters lacking apical–basal polarity, a hallmark of advanced epithelial cancers. Here, using live imaging of the developmental migration program of *Drosophila* primordial germ cells (PGCs), we show that cluster dispersal is accomplished by stabilizing and orienting migratory forces. PGCs utilize a G protein coupled receptor (GPCR), Tre1, to guide front-back migratory polarity radially from the cluster toward the endoderm. Posteriorly positioned myosin-dependent contractile forces pull on cell–cell contacts until cells release. Tre1 mutant cells migrate randomly with transient enrichment of the force machinery but fail to separate, indicating a temporal contractile force threshold for detachment. E-cadherin is retained on the cell surface during cell separation and augmenting cell–cell adhesion does not impede detachment. Notably, coordinated migration improves cluster dispersal efficiency by stabilizing cell–cell interfaces and facilitating symmetric pulling. We demonstrate that guidance of inherent migratory forces is sufficient to disperse cell clusters under physiological settings and present a paradigm for how such events could occur across development and disease.

[1] HHMI and Kimmel Center for Biology and Medicine of the Skirball Institute, Department of Cell Biology, New York University School of Medicine, New York, NY, USA. [2] Present address: Whitehead Institute for Biomedical Research and Department of Biology, Massachusetts Institute of Technology, Cambridge, MA, USA. ✉email: Benjamin.lin@nyulangone.org; Lehmann@wi.mit.edu

During development, external signals can shape embryonic architecture by instructing individual cells to detach from their neighbors and move to establish new tissues. Prominent examples include the migration of neural crest cells from the neural tube to form diverse tissues such as cartilage and the peripheral nervous system[1], as well as the movement of myogenic precursors from the dermomyotome towards limb buds[2]. Many pioneering cells have an epithelial origin and must delaminate to begin their journey. Live imaging has revealed that developmental delamination involves actomyosin dependent apical constriction and retraction[3–9], along with a concomitant loss of apical–basal polarity and/or adherens junctions that occurs prior to or after full detachment[5]. Notably, the integrity of the epithelial layer these cells emerge from is maintained[3–9]; thus, detaching cells generate forces against apically localized neighboring junctions[6,8]. Contractile forces are already apically positioned at junctions prior to delamination[3,6,8,9], most likely to stabilize junctions and maintain tissue cohesion[10]. Subsequent delamination is then driven by an increase in the magnitude of the junctional contractile force[3], the disassembly of adherens junctions[9], changes in amplitude or frequency of medial-apical contractile pulses which pull on junctions[6,8], or a combination of these processes.

Aside from an epithelium, individual cells can also detach from cell clusters lacking apical–basal polarity during developmental migration. Hematopoietic stem cells separate from intra-aortic hematopoietic clusters in mouse[11] and primordial germ cells (PGCs) disperse and migrate individually from clustered origins in *Drosophila*[12], mouse[13], zebrafish[14], and *Xenopus*[15]. In contrast to what is known about epithelial delamination, little is known about how actomyosin contractility contributes to cluster dispersal in vivo, despite relevance to understanding how cells detach from cell masses with disrupted apical–basal polarity, a hallmark of advanced epithelial cancers[16]. While one potential point of convergence between delamination and cluster separation is the modulation of cell–cell adhesion[12,14], a different set of challenges arises when a cell is tasked with separating from a dynamic cell cohort as opposed to a relatively static one. Clustered cell ensembles can be motile, generate traction forces on each other, and rapidly exchange neighbors, as observed during mammary branching morphogenesis[17] or adaptive immunity[18], creating an unstable substrate to detach from. Moreover, the stable apical–basal polarity present in an epithelium, which presents an existing template for contractile forces to mediate delamination, is now replaced by a dynamic front–back polarity in a cluster. This polarity can be labile and it is unclear how forces in this context are now directed toward separation.

The developmental migration of *Drosophila* PGCs is an excellent model to study how actomyosin contractility is deployed to separate cells from non-epithelial clusters. PGCs are a group of 30–40 cells born at the posterior of the embryo. During gastrulation, PGCs are swept into the interior of the embryo, where they reside as a tight cluster in a rosette configuration enveloped by the endoderm[19] (Fig. 1a). PGCs subsequently separate and individually transmigrate through the endoderm as it undergoes a developmentally programmed epithelial-to-mesenchymal transition (EMT)[12,20,21]. How cluster separation is achieved mechanistically remains elusive. Known autonomous proteins required for PGC cluster dispersal are the orphan G-protein-coupled receptor (GPCR), Tre1 (Fig. 1b) and its associated Gβγ subunit, consisting of Gβ13F and Gγ1, as well as the small Rho GTPase, RhoA[12,20,22], suggesting the involvement of contractile forces. However, actomyosin dynamics during PGC cluster dispersal remain uncharacterized and how Tre1 influences the spatiotemporal dynamics of these networks is unknown.

Here, utilizing the developmental dispersal of *Drosophila* PGCs, we demonstrate that cell detachment from a dynamic cell cohort in vivo is achieved by orienting and stabilizing posterior migratory forces towards cell contacts. These posterior contractile forces are positioned by Tre1, which establishes a population wide radial front–back migratory polarity away from the cluster. Successful separation is critically dependent upon the maintenance of myosin II dependent pulling on cell–cell adhesions, as transient pulling produced by random motility does not disrupt cluster integrity. E-cadherin, the chief cell–cell adhesion molecule linking PGCs, is retained on the cell surface during PGC separation and augmenting cell–cell adhesion does not prevent dispersal. Coordinated migration improves cluster dispersal efficiency by steadying cell–cell contacts and aligning opposing pulling forces. Overall, we provide a developmental mechanism describing autonomous cell separation from cell ensembles lacking apical–basal polarity, with implications for how detachment may occur inappropriately in disease[23].

## Results

**Characterization of PGC morphology and migration during PGC cluster dispersal.** We first sought to better characterize the morphological changes and cytoskeletal responses in individual PGCs during cluster dispersal. To this end, we created a transgenic line harboring a *nanos* promoter driven bicistronic cassette consisting of lifeact-tdTomato, to visualize F-actin, and tdKatushka2-CAAX, to visualize membranes. Two-photon imaging of this line allowed us to optically dissect individual PGC dynamics during developmental cluster dispersal. WT PGCs were initially tightly clustered within the endoderm cavity in a rosette configuration[12], with higher levels of F-actin assembly and/or contraction arising in cell interfaces abutting the rosette center (hereafter referred to as posterior) (Fig. 1a and Supplementary Movie 1). As development proceeded, the posterior area of PGCs, defined by the area between the nucleus and the posterior cortex, gradually decreased along with a concomitant increase in F-actin intensity, suggesting that the posterior was contracting (Fig. 1a, c and Supplementary Movie 2). The PGC anterior cortices were generally stable and exhibited brief pulses of F-actin assembly and disassembly while remaining pressed against the apical surface of the endoderm epithelium (Fig. 1a). Following the developmentally programmed EMT[21,24] in the surrounding endoderm, the posterior cortices of now transmigrating PGCs further contracted into bright foci, separating posterior PGC–PGC contacts until the foci remained as the primary tether between PGCs (Fig. 1a, cyan arrows). Subsequently, the foci were either tugged off of other PGCs and incorporated into trailing tails (Fig. 1a, shown in greater detail in Supplementary Fig. 1a and Supplementary Movie 3) or were snapped off and left behind in the endoderm cavity (Supplementary Fig. 1b and Supplementary Movie 3). Our live observations thus suggest that progressive polarized contraction occurs in WT PGCs prior to cell detachment.

To determine if polarized contractions are necessary for cluster dispersal, we next characterized PGC behavior in *tre1* embryos where clusters fail to disperse. As in WT clusters, *tre1* PGC clusters were enveloped by the surrounding endoderm. However, in contrast to the stereotypical rosettes we observed in WT clusters, *tre1* PGC clusters did not display any consistent organization (Fig. 1b). Strikingly, at a similar developmental timeframe (prior to endoderm EMT) when WT PGCs were immotile and undergoing polarized contraction, *tre1* PGCs continuously moved throughout the cluster (Fig. 1a, b, Supplementary Movies 1 and 4), suggesting that contractile machinery is available to drive movement when Tre1 is absent. This motility continued following endoderm EMT and *tre1* PGCs were ultimately unable to separate from each other (Fig. 1b). Of note, we did not observe concerted morphological changes and rarely

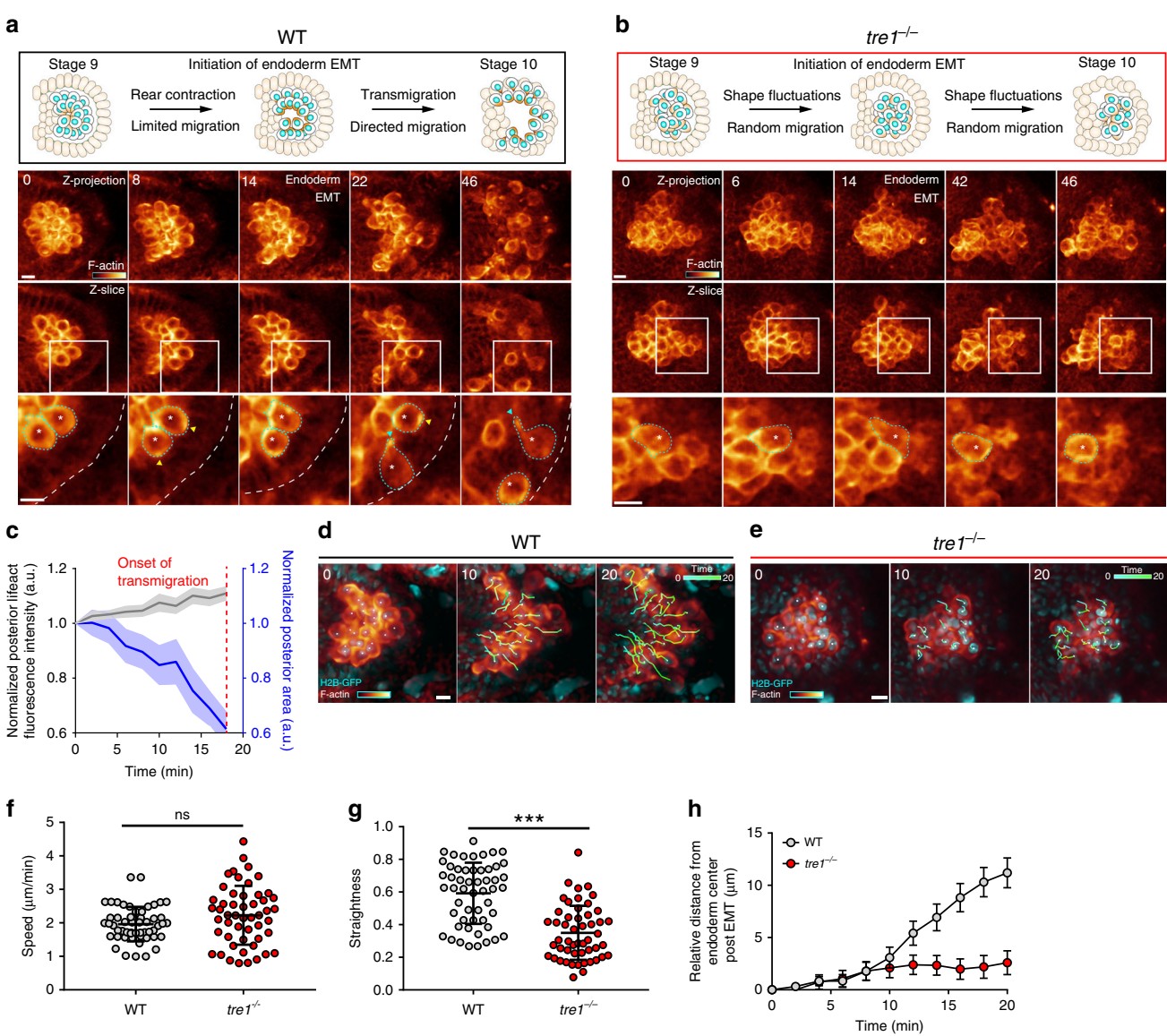

**Fig. 1 PGC morphology and migratory behavior during cluster dispersal.** Two-photon timelapse imaging of representative WT and *tre1*<sup>−/−</sup> PGC clusters from stage 9 to stage 10 of embryogenesis with a schematic (top) of morphological changes occurring in WT (**a**) and *tre1*<sup>−/−</sup> (**b**) PGC clusters and endoderm. Lifeact-tdTomato is presented in a pseudocolor and a corresponding color bar is shown in the first frame (intensity ranges: 979–13,000 (**a**), 184–4000 (**b**)). The top row of images is a maximum intensity projection, while the next row is a single Z slice, with a white rectangle indicating the region expanded in the bottom row. Blue arrows indicate F-actin foci at the posterior while yellow arrows highlight F-actin pulses at the anterior. White asterisk and cyan outlines track individual PGCs shown in Supplementary movies 2 (**a**) and 4 (**b**). Representative images are from n = 6 embryos in (**a**) and n = 5 embryos in (**b**). Times indicated are in minutes. Scale bars, 10 μm. **c** Quantification of normalized posterior lifeact-tdTomato intensity and area over time. n = 25 PGCs from 6 embryos. Error bars show SEM. Two-photon timelapse imaging of representative WT (**d**) or *tre1*<sup>−/−</sup> (**e**) PGC clusters expressing life act-tdTomato and H2B-GFP after the onset of endoderm EMT. Lifeact-tdTomato is presented in a pseudocolor and a corresponding color bar is shown in the first frame (intensity ranges: 60–15693 (**d**), 36–3075 (**e**)). White squares mark nuclei being tracked over time. Tracks are shown as a temporal gradient from cyan to green. Representative images are from n = 4 embryos in (**d**) and (**e**). Times are in minutes. Scale bars, 10 μm. Quantification of speed (**f**), straightness (**g**), and distance from the endoderm center over time (**h**) of WT (n = 53 PGCs from 4 embryos) and *tre1*<sup>−/−</sup> (n = 51 PGCs from 4 embryos) PGCs after endoderm EMT. Data are presented as mean values ± SD in (**f**, **g**). Error bars are SEM in (**h**). Statistical comparisons are from a Mann–Whitney test in (**f**, **g**) with p < 0.0001 in (**g**). ***p < 0.001. Source data in (**c**, **f**, **g** and **h**) are provided as a Source Data file.

observed F-actin foci in *tre1* PGCs, suggesting they were unable to generate sufficient contractile forces for detachment (Fig. 1b). Co-expression of a nuclear marker allowed us to quantify WT and *tre1* PGC migration following endoderm EMT (Fig. 1d–h). WT and *tre1* PGCs migrated with a similar speed, suggesting motility was not compromised (Fig. 1f). However, while WT PGCs took a fairly linear path through the endoderm, *tre1* PGCs moved randomly and not as extensively (Fig. 1g, h and Supplementary Movie 5). Overall, our results suggest that cluster dispersal

requires a Tre1 dependent shift of contractile signaling from motility towards cell–cell separation prior to endoderm EMT and directed migration through the endoderm following EMT.

**RhoA signaling during PGC cluster dispersal**. To determine if contractile forces are present during cell detachment, we next sought to visualize contractile signaling during cluster dispersal. The small Rho GTPase RhoA is a central regulator of contractility

in morphogenesis[25], but its spatial activity profile in PGC clusters has not been determined. To visualize RhoA activity live in WT and *tre1* PGC clusters, we utilized transgenic flies expressing a *nanos* promoter driven bicistronic transgene consisting of a previously described Anillin RhoA-GTP binding domain (RBD)[26–28] (AH–PH domain from Anillin) fused to tdTomato which enriches along membranes where RhoA is active, along with a membrane marker (tdKatushka2-CAAX) (Supplementary Fig. 2a, b). In WT clusters, we observed a prominent enrichment of signal from the Anillin-RBD in the cluster center, which sharply decayed with increasing distance (Fig. 2a, c). This enrichment overlapped with PGC posterior membranes (Fig. 2a), supporting our previous observations of posterior contraction (Fig. 1a, c), and was reduced when overexpressing a dominant negative RhoA relative to WT RhoA (Supplementary Fig. 2c, e, f). Overexpression of constitutively active RhoA did not perturb this distribution (Supplementary Fig. 2d, f), suggesting an ability to self-organize polarity. The Anillin-RBD was also locally enriched along a subset of membranes in *tre1* PGC clusters; however, these Anillin-RBD laden membranes were present throughout the cluster, likely marking contractile signaling at the rear of moving *tre1* PGCs. Overall, the Anillin-RBD was more broadly distributed throughout *tre1* PGC clusters while retaining a weaker enrichment in the cluster center as compared to WT (Fig. 2b, c). Thus, as opposed to our previous interpretation that Tre1 generates RhoA polarity[12,22], Tre1 regulates the orientation of an intrinsic RhoA polarity.

We confirmed these Anillin-RBD observations by imaging the distribution of two well characterized downstream RhoA effectors, Dia and ROCK, which nucleate linear F-actin and activate myosin II[25], respectively (Supplementary Fig. 2a). Overexpression of GFP-Dia-RBD or GFP-Rock along with the Anillin-RBD confirmed that they co-localized and were indeed enriched in the center of WT clusters along the posterior of PGCs (Fig. 2d–g). In *tre1* PGC clusters, GFP-ROCK was enriched in discrete regions throughout the cluster, in accord with our Anillin-RBD observations (Fig. 2h). This led to a uniform GFP-ROCK distribution when averaging across many clusters (Fig. 2i). Lastly, we also assessed whether Cdc42 and Rac, other small Rho GTPases prominently involved in regulating cytoskeletal dynamics, displayed polarized activity in PGC clusters (Supplementary Fig. 3). GFP-tagged Cdc42-GTP and Rac-GTP binding domains[29] remained cytoplasmic and were uniformly distributed in WT clusters, suggesting that Rac and Cdc42 activities are unpolarized (Supplementary Fig. 3). However, we are unable to rule out subtle differences in localization due to saturation. Taken together, these findings suggest that RhoA signaling is active at sites of cell–cell separation.

**PGC detachment requires a stable myosin II polarity**. Because the contractile forces generated by randomly moving *tre1* PGCs are not sufficient for cluster dispersal, we hypothesized that higher levels of contractility are required for cell–cell detachment. With the Anillin-RBD, we were unable to distinguish differences in the magnitude of RhoA signaling utilized by WT and *tre1* PGCs due to the close apposition of membranes within clusters. RhoA signaling ultimately generates contractility by relieving autoinhibition of ROCK, which subsequently activates myosin II by phosphorylating the myosin II regulatory light chain (RLC), leading to its assembly into bipolar minifilaments which contract actin networks[25]. Therefore, to more accurately assess contractile signaling levels, we visualized the distribution of myosin II in PGC clusters. We utilized a GFP-tagged myosin II RLC transgene under endogenous regulation as a readout for myosin II activity (Fig. 3), as has been done previously[30,31]. As expected, myosin II was clearly polarized at the posterior of WT PGCs within the

cluster and displayed a consistent orientation in relation to the cluster center (Fig. 3a, d). In *tre1* PGC clusters, myosin II was also polarized within individual PGCs, but the orientation of myosin II polarity in *tre1* PGCs was completely random with respect to the cluster center (Fig. 3b, d), as would be expected for randomly moving cells. Surprisingly, in contrast to our expectation of differences in myosin II enrichment between WT and *tre1* PGCs, myosin II was polarized to the same extent in individual WT and *tre1* PGCs (Fig. 3c). Thus, at any given time point prior to endoderm EMT, the only feature that distinguishes WT from *tre1* PGCs is the orientation of contractile signaling.

Our prior observations of progressive anisotropic contraction during PGC separation (Fig. 1c) suggest that myosin II could be stably polarized at the posterior of WT PGCs. Moreover, the concomitant reduction in cell–cell contact during contraction further suggests that polarity must be maintained for a sufficient period of time for full detachment. Differences in the stability of myosin II polarity between WT and *tre1* PGCs could thus explain why cluster dispersal fails in the absence of Tre1. To explore this possibility, we assessed the stability of myosin II polarity in PGCs by performing timelapse imaging of PGC clusters expressing myosin II RLC GFP and lifeact-tdTomato (Fig. 3e–h), first focusing on cluster wide patterns of myosin II. WT PGCs exhibited a stable, cluster level myosin II polarity for at least 18 min prior to endoderm EMT, suggesting a striking coordination amongst the population whereby an oriented myosin II polarity was maintained within individual PGCs (Fig. 3e, g, h, Supplementary Fig. 4a, and Supplementary Movie 6). This cluster level polarity was also apparent when the sole source of myosin II RLC was from the myosin II RLC-GFP transgene, ruling out artifacts from overexpression (Supplementary Fig. 4c, d). In *tre1* PGC clusters, we observed local regions of myosin II enrichment scattered throughout the cluster, corresponding to individual PGC myosin II polarity, which rapidly changed position over time, suggesting that *tre1* PGCs reorient myosin II polarity and/or lose polarity and repolarize elsewhere during random movement (Fig. 3f, Supplementary Fig. 4b, and Supplementary Movie 7). Due to the dynamic nature of myosin II polarity in *tre1* PGCs, there was no discernable cluster wide myosin II organization at any time point after averaging many clusters (Fig. 3f–h). These results suggest that a stable, oriented myosin II polarity is necessary for cluster dispersal.

Although cluster level measurements can suggest how individual PGCs behave, a more definitive characterization requires tracking of individual cells over time. To enable single-cell tracking, we improved the spatial–temporal resolution of our live imaging experiments by utilizing a recently developed myosin II RLC 3xGFP transgene under its native promoter[32]. This enabled us to track myosin II polarity in individual WT PGCs at 30-s intervals within PGC clusters and confirmed that polarity was strikingly stable (Fig. 3i, j and Supplementary Movie 8). We did not have sufficient temporal resolution to rule out pulsatile accumulation of myosin II, as is frequently observed during apical constriction in epithelial[6,8,25,33], but we did not observe sustained changes in myosin II levels, suggesting that a stable myosin II polarity promotes PGC detachment through the temporal integration of a constant contractile force (Fig. 3i, j).

**Migratory forces are utilized to separate PGCs**. Even with the improved myosin II RLC 3xGFP transgene, tracking individual myosin II polarity in *tre1* PGC clusters was exceedingly difficult due to the constant 3D movement of ~30 closely positioned cells. To overcome this technical challenge, we transplanted small numbers of WT or *tre1* PGCs expressing lifeact-tdTomato and myosin II RLC 3xGFP into the posterior pole of early embryos genetically

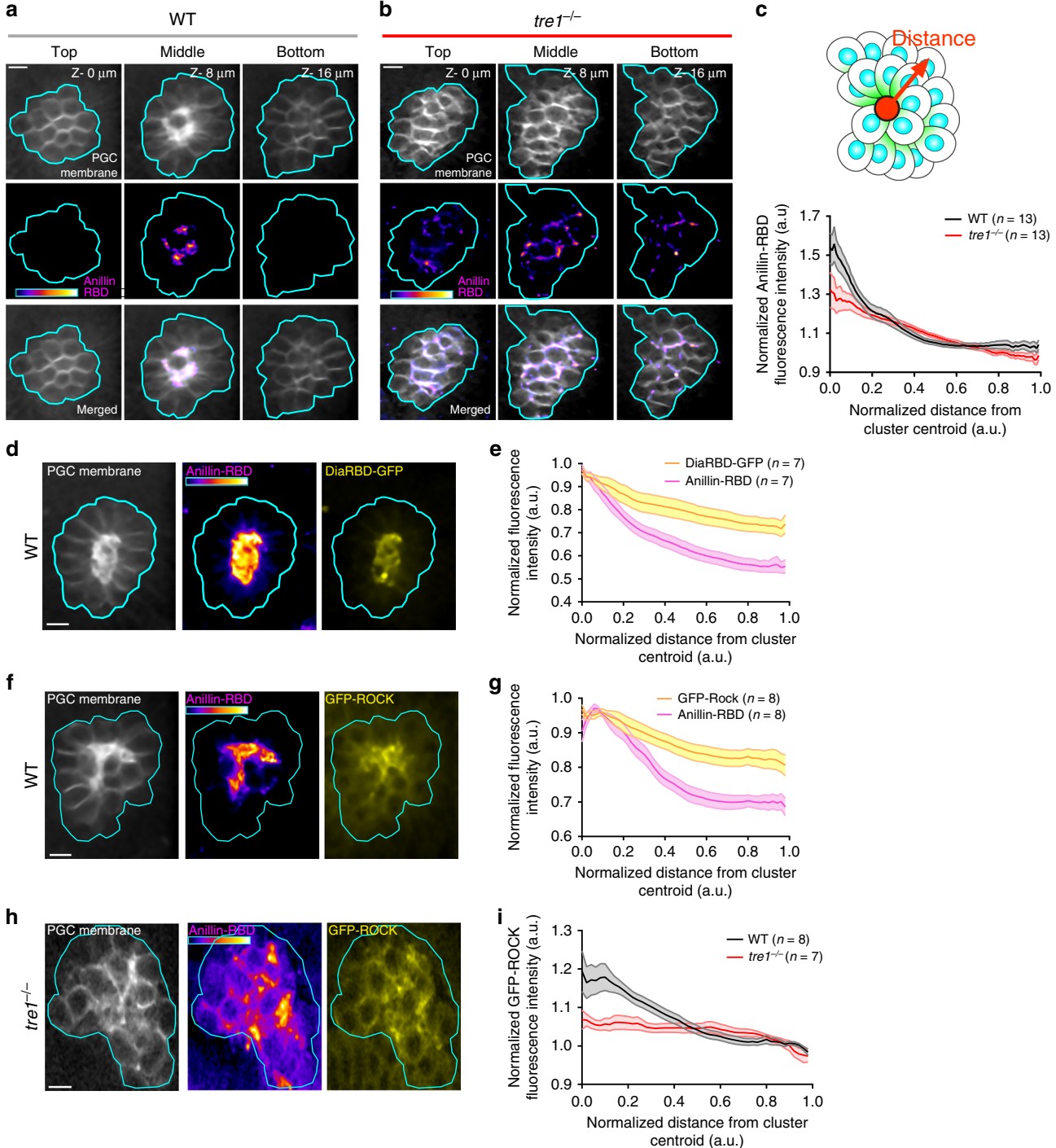

**Fig. 2 The RhoA signaling pathway is active at sites of PGC separation.** Two photon imaging of a representative WT (**a**) or *tre1*−/− (**b**) PGC cluster expressing Anillin-RBD. Three different Z slices at the indicated depth relative to the first slice are shown. The Anillin-RBD is pseudocolored with the color bar within the image (intensity range: 664–2175 in (**a**, **b**). Representative images are from *n* = 16 embryos in (**a**) and *n* = 14 embryos in (**b**). **c** Quantification of normalized Anillin-RBD intensity relative to distance from the cluster centroid in WT and *tre1*−/− PGC clusters. Number of clusters are indicated. Shaded region represents SEM. Two photon imaging of a single Z slice from a representative WT PGC cluster expressing the Anillin-RBD and Dia-RhoA-GTP Binding Domain (RBD)-GFP (**d**) or GFP-ROCK (**f**). The Anillin-RBD is pseudocolored with the color bar within the image (intensity ranges-1747–3903 in (**d**), 1429–3820 in (**f**). Representative images are from *n* = 7 embryos in d and *n* = 8 embryos in (**f**). Quantification of normalized fluorescence intensity of Anillin-RBD and DiaRBD-GFP (**e**) or GFP-ROCK (**g**) in PGC clusters relative to the distance from the cluster centroid. Values are normalized to max intensity. Shaded region represents SEM. **h** Two photon imaging of a single Z slice from a representative *tre1*−/− PGC cluster expressing Anillin-RBD and GFP-ROCK. Anillin-RBD is pseudocolored with the indicated color bar in the image (Intensity range = 540–2346). Representative images are from *n* = 7 embryos in (**h**, **i**) Quantification of normalized fluorescence intensity of GFP-ROCK in WT and *tre1*−/− PGC clusters relative to the distance from the cluster centroid. Values are normalized to background in (**c**, **i**). Number of clusters are indicated. Shaded region represents SEM. All scale bars, 10 µm. Source data in (**c**, **e**, **g** and **i**) are provided as a Source Data file.

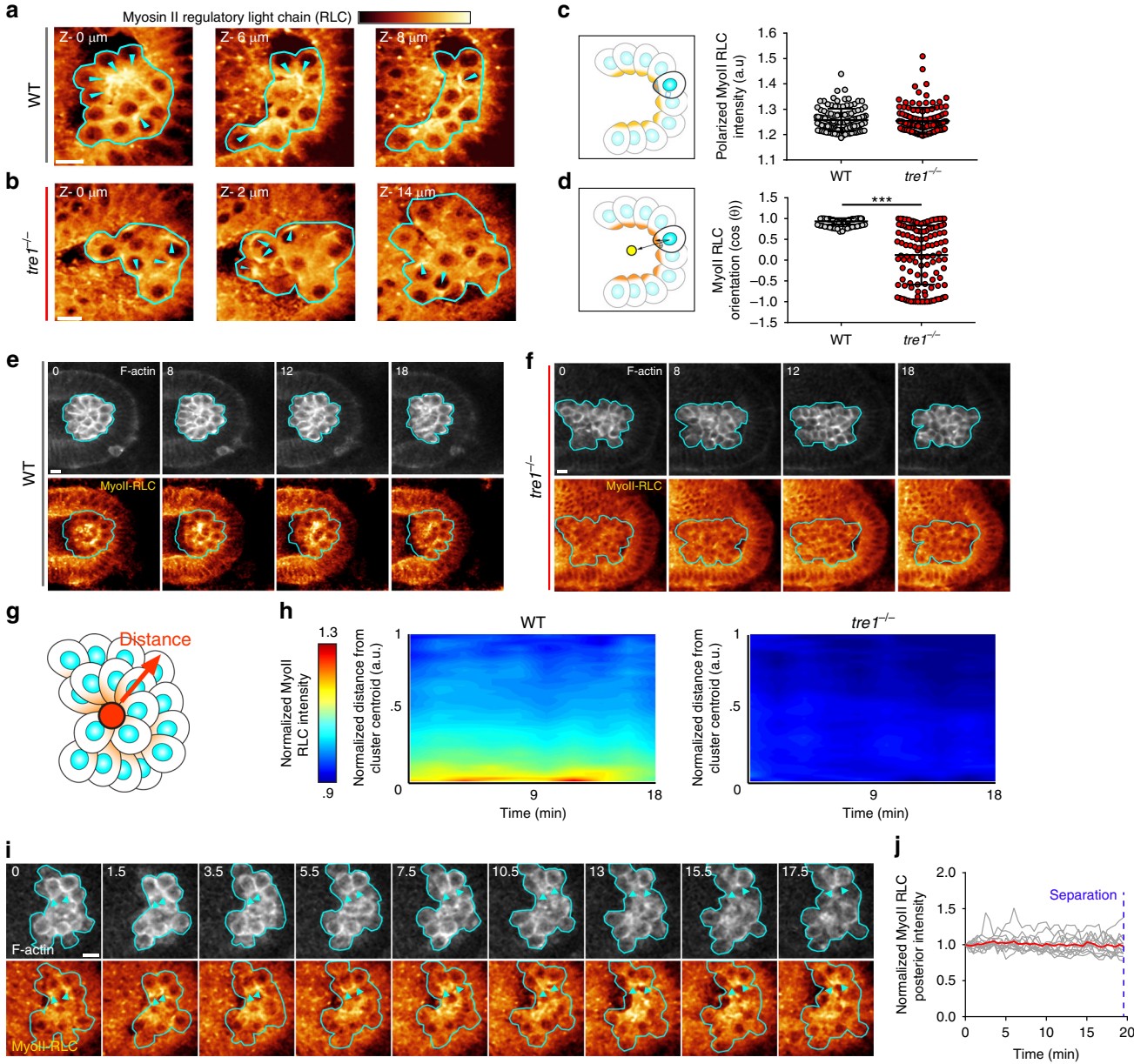

**Fig. 3 Myosin II polarity is collectively oriented and remains persistent during separation. a, b** Two-photon imaging of myosin II-RLC GFP in a representative WT (**a**) or *tre1*$^{-/-}$ (**b**) cluster. Myosin II-RLC is pseudocolored with the indicated color bar on top (intensity ranges: 2319–4005 (**a**) and 2203–3803 (**b**)). Different Z slices with indicated depth relative to the first slice are shown. Cyan arrows indicate regions of myosin II-RLC accumulation. Representative images are from *n* = 15 embryos in (**a**) and *n* = 13 embryos in (**b**). **c, d** Quantification of polarized myosin II RLC intensity (**c**) and myosin II RLC orientation (**d**) in WT (*n* = 123 from 15 embryos) or *tre1*$^{-/-}$ (*n* = 127 from 13 embryos) PGCs. Data are presented as mean values ± SD. Statistical comparisons are from a Mann–Whitney test in (**d**) with *p* < 0.0001. ***$p$ < 0.001. Two-photon timelapse imaging of a representative WT (**e**) or *tre1*$^{-/-}$ (**f**) PGC cluster expressing lifeact-tdTomato and myosin II RLC-GFP for 18 min prior to endoderm EMT. Myosin II-RLC is pseudocolored with the indicated color bar above (**a**) (intensity ranges: 1072–2171 (**e**), 3198–7726 (**f**)). A single Z slice is shown. Representative images are from *n* = 6 embryos in (**e** and **f**). **g** Schematic of how intensity vs. distance from cluster centroid is calculated. **h** Quantification of the average myosin II RLC intensity vs. distance from the cluster centroid over time in WT (*n* = 6 embryos) and *tre1*$^{-/-}$ (*n* = 6) PGC clusters. **i** Two photon timelapse imaging of a representative WT PGC cluster expressing lifeact-tdTomato and myosin II RLC-3xGFP over time. Myosin II-RLC is pseudocolored with the indicated color bar on top of (**a**) (intensity range: 2560–5624). A single Z slice is shown. Cyan arrows indicate regions of myosin II RLC accumulation from the same pair of cells. Representative images are from *n* = 5 embryos. **j** Quantification of myosin II RLC-3xGFP intensity at the posterior over time. Grey curves show single cells while the red curve indicates the mean. *n* = 13 PGCs from 5 embryos. All scale bars, 10 μm. All times are in minutes. Source data in (**c, d, h** and **j**) are provided as a Source Data file.

devoid of PGCs (Fig. 4a). These transplanted PGCs spread out and remained attached to the underlying somatic cells and were subsequently carried into the interior of the embryo during gastrulation, mimicking normal PGC development. This simplified context

allowed us to track migration and myosin II polarity with high fidelity in individual PGCs and also provided an opportunity to test our hypothesis for the mechanism underlying PGC separation. We reasoned that if WT PGCs generate contractile signaling purely for

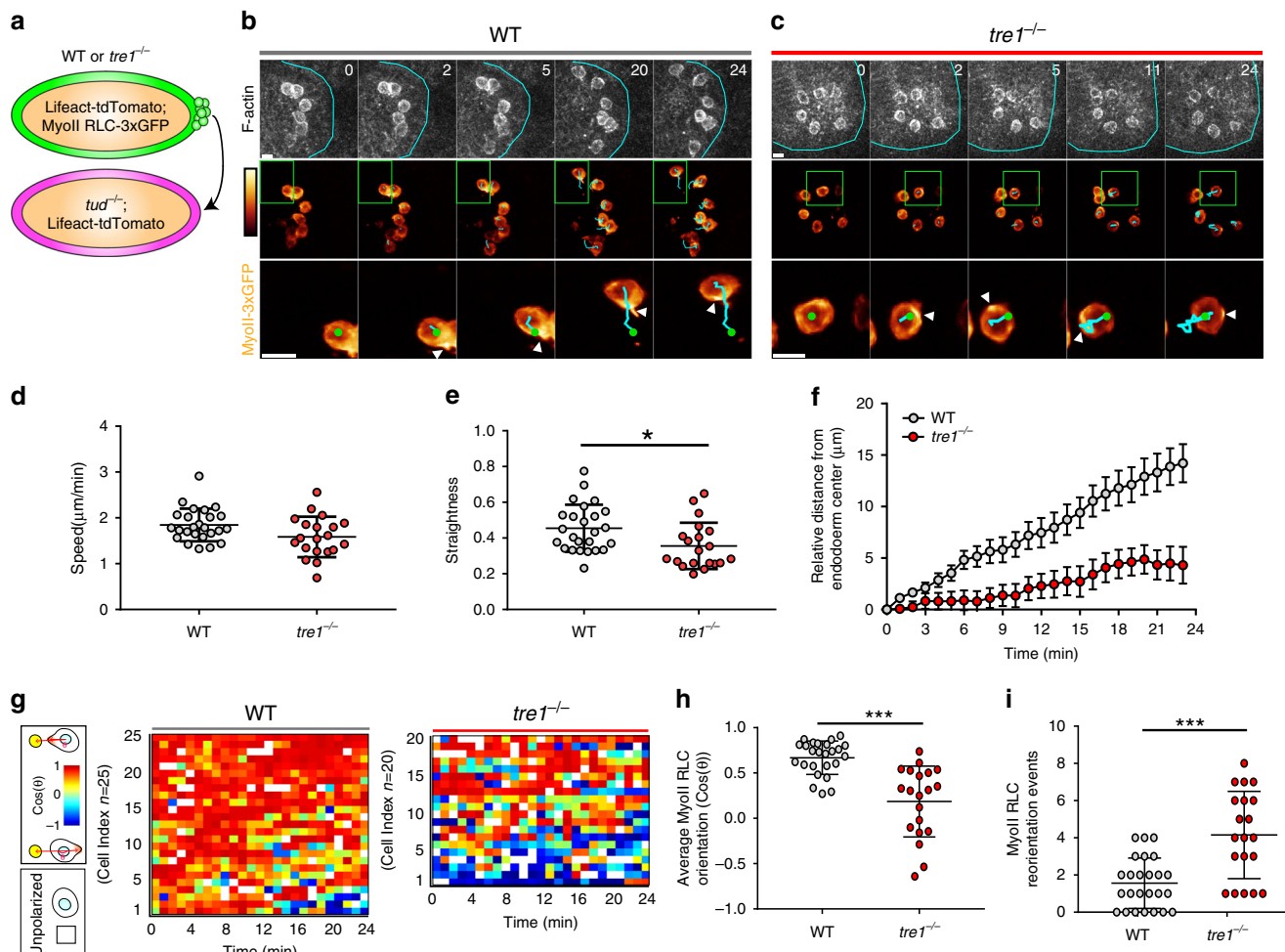

**Fig. 4 Migration machinery is stabilized and oriented to drive PGC detachment. a** Schematic describing transplantation of PGCs with the indicated genotypes into embryos genetically devoid of PGCs. Two photon timelapse imaging of representative transplanted WT (**b**) and $tre1^{-/-}$ (**c**) PGCs. Cyan outlines the endoderm in the top set of panels. Myosin II RLC-3xGFP is pseudocolored with color bar indicated on the left (intensity ranges: 793–3616 (**b**), 801–3509 (**c**)). Green box indicates expanded region on the bottom row of images. Green circles mark nuclei, cyan lines indicate cell tracks, and white arrows highlight regions of myosin II RLC-3xGFP accumulation. Representative images are from $n = 10$ embryos in (**b**) and $n = 14$ embryos in (**c**). Times are in minutes. Scale bars, 10 µm. Quantification of speed (**d**), straightness (**e**), and relative distance from the endoderm center (**f**) for transplanted WT ($n = 25$ PGCs from 10 embryos) and $tre1^{-/-}$ ($n = 20$ PGCs from 14 embryos) PGCs. Data are presented as mean values ± SD in (**d**, **e**). Error bars are SEM in (**f**). Statistical comparisons are from a Mann–Whitney test in (**e**) with $p = 0.0149$. *$p = 0.05$. **g** Quantification of myosin II RLC 3xGFP polarity angles in single WT or $tre1^{-/-}$ PGCs. Each row shows the angles of a single cell over time. Polarity angles are pseudocolored as indicated on the left with the yellow circle representing the endoderm center, while a white box indicates when no polarity was detected. Quantification of average Myosin II RLC orientation (**h**) and Myosin II RLC reorientation events (**i**) (Shift > 60°) in WT ($n = 25$ PGCs from 10 embryos) and $tre1^{-/-}$ ($n = 20$ PGCs from 14 embryos) PGCs. Data are presented as mean values +/− SD. Statistical comparisons are from a Mann–Whitney test with $p < 0.0001$ in h and $p = 0.0002$ in (**i**). ***$p < 0.001$. Source data in (**d–i**) are provided as a Source Data file.

cell–cell detachment, then they should remain stationary but polarized prior to endoderm EMT. Strikingly however, transplanted WT PGCs, now unconstrained by other PGCs within the endoderm cavity, directionally migrated toward the periphery of the pre-EMT endoderm without prematurely crossing (Fig. 4b, d–f, Supplementary Movies 9 and 10), suggesting that WT PGCs utilize migratory forces to separate rather than a distinct contractile program. This notion is supported by the equivalent myosin II polarity in WT and *tre1* PGCs (Fig. 3c) and precocious cluster dispersal when the endoderm prematurely breaks down in *crumbs* mutants[21]. Similar to what we observed in clusters (Fig. 1e–h), transplanted *tre1* PGCs moved randomly, frequently switching directions and retracing previous paths (Fig. 4c, d–f, Supplementary Movies 9 and 11). Myosin II was stably polarized at the rear of transplanted WT PGCs and remained oriented toward the center of the endoderm during outward migration (Fig. 4g–h). In contrast, although myosin II was

also polarized at the rear of transplanted *tre1* PGCs, transplanted *tre1* PGCs frequently shifted their myosin II orientation during random movement and exhibited significantly more reorientation events (shift in myosin II polarity >60°) than WT PGCs (Fig. 4g–i). We conclude that PGC cluster dispersal is driven by the stabilization of front–back migratory polarity which positions migratory forces to constrict and pull the cell rear until cell release.

**Myosin II is necessary for cluster dispersal and can redirect migration**. To determine if contractile forces are indeed necessary for PGC separation, we sought to inhibit myosin II function in PGCs prior to cluster dispersal. We capitalized on a GFP degradation system, termed degradFP, which has previously been utilized to inhibit myosin II by degrading myosin II RLC GFP[34]. We used MatTub-Gal4-VP16 to drive the degradFP system maternally in a genetic background where all maternally supplied

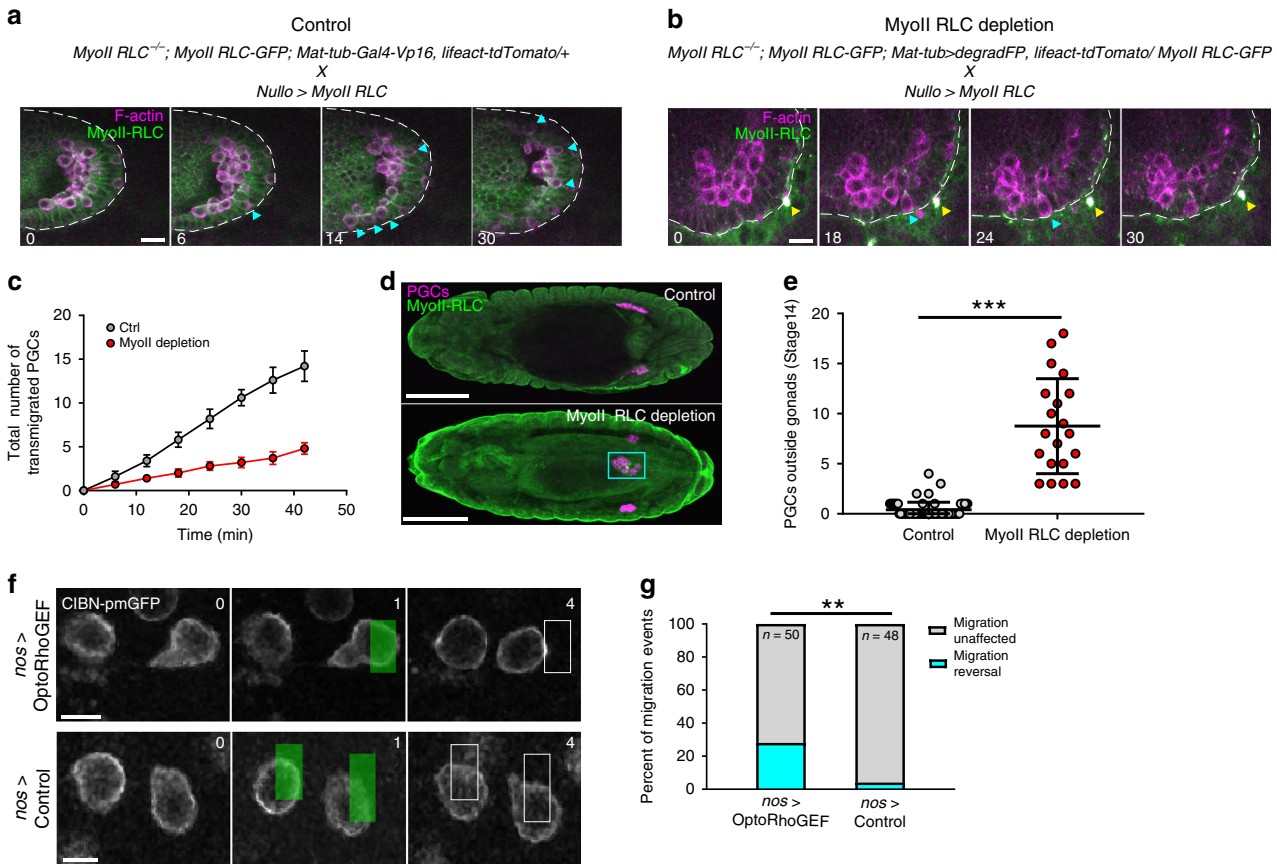

**Fig. 5 Myosin II is necessary for PGC dispersal and its local accumulation can redirect migration. a, b** Two photon timelapse imaging of a representative control and Myosin II RLC depleted PGC cluster for 30 min after endoderm EMT. A single Z slice is shown. White dotted lines trace the boundary of the endoderm. Cyan arrows indicate PGCs which have transmigrated. Yellow arrow indicates a myosin II RLC-GFP inclusion body, which has been observed previously after degradation. Representative images are from $n = 5$ embryos in (**a**) and $n = 10$ embryos in (**b**). Times are in minutes. Scale bars, 10 μm. **c** Quantification of the total number of PGCs which have transmigrated over time under the indicated conditions. $n = 10$ embryos for myoII depletion and $n = 5$ embryos for control. Error bars are SEM. **d** Representative confocal images of immunostained stage 14 embryos under the indicated conditions. Representative images are from $n = 65$ embryos in control and $n = 20$ embryos in myoII depletion. Scale bars, 100 μm. **e** Quantification of the number of PGCs outside the gonad at stage 14 of embryogenesis. $n = 65$ control embryos and $n = 20$ myoII RLC depletion embryos. Data are presented as mean values $+/-$ SD. Statistical comparisons are from a Mann–Whitney test with $p < 0.0001$. ***$p < 0.001$. **f** Two photon timelapse imaging of representative PGCs expressing the OptoRhoGEF (RhoGEF2-CRY2-mCherry, CIBn-pmGFP) system (top) or control (CIBnpmGFP) (bottom) migrating to the anterior (right) at stage 11 of embryogenesis. A single Z slice is shown. Green box indicates region which was pulsed with 950 nm light. The white rectangle in the following frame indicates the position of the pulse at time $t = 1$. Representative images are from $n = 9$ embryos in OptoRhoGEF and $n = 10$ embryos in control. Times are in minutes. Scale bars, 10 μm. **g** Quantification of the percent of migration events which occur after a blue light pulse under the indicated condition. Number of pulses are indicated. $n = 27$ PGCs from 9 embryos for OptoRhoGEF and $n = 25$ PGCS from 10 embryos for control. Statistical comparisons are from a Fisher's exact test with $p = 0.0019$. **$p < 0.01$. Source data from (**c**, **e** and **g**) are provided as a Source Data file.

myosin II RLC is GFP tagged. To target the system to the posterior pole where PGCs are specified, we developed a hybrid-3′ UTR, *nos*TCE-*pgc*-3′UTR, that consists of a fusion between the *nanos* (*nos*) translational control element (first 100 bp of the *nos* 3′UTR) and the *polar granule component* (*pgc*) 3′UTR. Together these elements prevent translation during oogenesis and allow subsequent enrichment at the posterior pole where PGCs are specified (Supplementary Fig. 5, see "Methods" for description). To rescue potential defects in somatic development due to leakage of degradFP from the posterior of the embryo, we also over-expressed untagged, and therefore degradFP resistant myosin II RLC utilizing an early soma specific driver, *nullo*-Gal4. Myosin II depletion significantly decreased the rate at which PGCs separated from clusters (Fig. 5a–c, Supplementary Movies 12 and 13) and caused many PGCs to remain clustered in the endoderm later in development at stage 14, when PGC migration is complete (Fig. 5d, e). We speculate that we did not achieve complete

inhibition of cluster dispersal because of incomplete degradation or partial rescue of myosin II function by zygotic expression of untagged myosin II RLC in PGCs. Taken together, these results indicate that myosin II dependent contractile forces are necessary for PGC cluster dispersal.

Hallmarks of many directionally migrating cells are the reciprocal localization of active Rac1/Cdc42 and RhoA, which establish and reinforce "front" and "back" signaling modules, respectively[35]. In agreement with this paradigm, myosin II is enriched at the rear of WT PGCs within clusters (Fig. 3e, h). However, Cdc42-GTP and Rac1-GTP binding domains are not polarized at the front (Supplementary Fig. 3) and overexpression of dominant negative or constitutively active Rac1 or Cdc42 does not prevent cluster dispersal[20], suggesting that PGC directionality could be conveyed by an alternative mechanism. We thus wondered whether the accumulation of myosin II would be sufficient to set the direction of PGC migration. To test this, we

employed an optogenetic system to locally active RhoA under defined regions of blue light illumination, which was previously shown to rapidly recruit myosin II and induce apical constriction in *Drosophila* embryonic epithelial cells[36]. We were unable to locally activate RhoA within PGC clusters due to their depth within embryos, therefore we actuated the system during stage 11 of embryogenesis when PGCs have separated from clusters and migrate closer to the dorsal surface of the embryo within the mesoderm. At this developmental stage, PGCs directionally migrate toward the anterior of the embryos, allowing us to test whether local RhoA activation at the leading edge is sufficient to redirect migration towards the posterior. Local RhoA activation caused a striking migration reversal in 28% of the blue light pulses we delivered to the leading edge, whereas only 4% of control pulses produced reversion in controls (Fig. 5f–g, Supplementary Movies 14 and 15). The relatively low reversal rates we observed in our experiments are likely due to the depth of PGCs at this developmental stage (~50–80 μm) and the decreased 2P actuation efficiency of CRY2-CIBN systems[37]. Our results suggest that local RhoA activation and likely subsequent myosin II recruitment can specify the direction of PGC migration.

**E-cadherin remains on the cell surface during dispersal and increased cell–cell adhesion does not inhibit separation**. Cell–cell separation events are frequently linked to a loss of cell–cell adhesion through downregulation of cadherins[38]. However, recent work has challenged this dogma[39] and it remains unclear whether a reduction in cell–cell adhesion is required for detachment. Thus, we sought to determine whether there were alterations in cell–cell adhesion proteins during developmental cluster dispersal. E-cadherin is the chief adhesive molecule which links PGCs to each other and the underlying somatic cells at this developmental stage[40] and we have previously shown that it becomes enriched at the posterior of WT PGCs prior to dispersal in fixed embryos[12]. In *tre1* PGCs, E-cadherin remains uniform, suggesting that Tre1 plays a role in redistributing E-cadherin. To confirm this enrichment and to monitor E-cadherin levels during and after separation in the same cells, we transplanted PGCs expressing an E-cadherin 3xGFP knock-in[32] and lifeact-tdTomato into the posterior pole of WT embryos solely expressing lifeact-tdTomato (Fig. 6a–c). This allowed us to track E-cadherin in single PGCs within a native context. In WT PGCs, E-cadherin was enriched at tail regions during cell separation, as we had previously shown[12] (Fig. 6a and Supplementary Movie 16). Following the appearance of an F-actin focus and complete cell detachment, E-cadherin remained enriched at the posterior of directionally migrating WT PGCs (Fig. 6a). We also observed transient, posterior E-cadherin enrichment in E-cadherin-3xGFP expressing *tre1* PGCs transplanted into *tre1* embryos during brief periods of more directed migration (Fig. 6b and Supplementary Movie 17), indicating that rearward E-cadherin enrichment does not require Tre1. Overall, the E-cadherin membrane to cytoplasm ratio was similar in WT and *tre1* PGCs and remained constant, suggesting that developmental cluster dispersal does not require detectable alterations in cell–cell adhesion (Fig. 6c). We confirmed this result by co-staining E-cadherin with an early endosome marker, Rab5 and late endosome marker, Rab7 before and after endoderm EMT in WT and *tre1* PGCs (Supplementary Fig. 6). Because cluster dispersal occurs within ~60 min and E-cadherin cleaving matrix metalloproteases are not expressed in the endoderm or PGCs at this developmental stage[41], alterations in PGC E-cadherin availability are likely only driven by surface retention. E-cadherin remained on the membranes of WT and *tre1* PGCs and did not show any changes in co-localization with Rab5 or Rab7, reinforcing the notion that E-cadherin is retained at the cell surface during cluster dispersal (Supplementary Fig. 6). Thus, although a decrease in E-cadherin levels is sufficient to disband *tre1* PGC clusters[12], overt E-cadherin regulation appears to be dispensable for WT cluster dispersal.

Because cadherin overexpression can delay or prevent cell separation from epithelial tissues[9], we asked whether PGCs challenged with increasing levels of adhesion would be able to disperse. To increase cell–cell adhesion in PGCs prior to cluster dispersal, we created UAS driven *E-cadherin* and *Neuroglian* transgenes coupled to the *nos*TCE-*pgc* 3′UTR described above. We chose these molecules because they represent calcium dependent (E-cadherin) and independent (Neuroglian) means to increase adhesion, are sufficient to ectopically adhere insect S2 cells, and associate with different cytoplasmic effectors[42,43], thus informing us more generally how altering adhesion affects PGC dispersal. Interestingly, overexpressing E-cadherin or Neuroglian did not delay PGC cluster dispersal (Fig. 6d–g, Supplementary Movies 18 and 19), suggesting that the contractile forces generated by migrating PGCs are sufficient to overcome increased adhesion. We confirmed that the overexpressed E-cadherin and Neuroglian were functional by driving their expression in *tre1* PGCs. Overexpression of either molecule caused *tre1* PGCs to increase their surface contact with each other through flattening of the stereotypical spherical PGC morphology (Fig. 6h–j), suggesting an increased level of adhesion. We conclude that a directed migration-based dispersal mechanism is robust to increases in cell–cell adhesion levels.

**Efficient PGC cluster dispersal requires cell coordination**. PGC cluster dispersal is a remarkably coordinated event where every PGC is directed to migrate away from the cluster (Fig. 1a). We wondered whether this coordination facilitates dispersal. To alter the levels of coordination between PGCs within a given cluster, we created chimeric clusters by transplanting WT PGCs into *tre1* PGC clusters. Altering the number of transplanted PGCs allowed us to favor the formation of chimeric clusters with distinct compositions—small numbers of transplanted PGCs tended to produce individual transplanted PGCs surrounded by host PGCs, while large numbers of transplanted PGCs created groups surrounded by host PGCs (Fig. 7a–d). Thus, we were able to assess whether increasing levels of coordination (individual vs. group) would correlate with successful separation. Strikingly, individual WT PGCs residing in the interior of chimeric clusters contacting ≥3 *tre1* PGCs had an approximately eightfold reduction in transmigration success following endoderm EMT (Fig. 7a, e), while individual WT PGCs at the cluster edge, contacting ≤ 2 *tre1* PGCs, were only slightly less successful than controls (Fig. 7a, b, e and Supplementary Movie 20). The WT PGCs in the cluster interior were not shielded from sensing the Tre1 guidance cue (Fig. 7f); however, their migration speed was significantly reduced (Fig. 7g), suggesting that the random movement of *tre1* PGCs exert counterproductive traction forces on WT PGCs. Defects in WT PGC transmigration from the interior of *tre1* clusters could alternatively result from aberrant adhesion between WT and *tre1* PGCs, leading to inefficient motility. However, we did not find significant differences in the membrane levels of the chief PGC–PGC adhesive molecule, E-cadherin, between WT and *tre1* PGCs during live imaging of endogenously tagged E-cadherin (Fig. 6c). Groups of WT PGCs (≥3 WT PGCs) had reduced contact with *tre1* PGCs and migrated outwards concurrently (Fig. 7c), detaching from *tre1* PGC clusters with a ~1.4-fold reduction in frequency compared to controls (Fig. 7c–e and Supplementary Movie 21). These results suggest that increasing cell–cell coordination improves cluster dispersal efficiency.

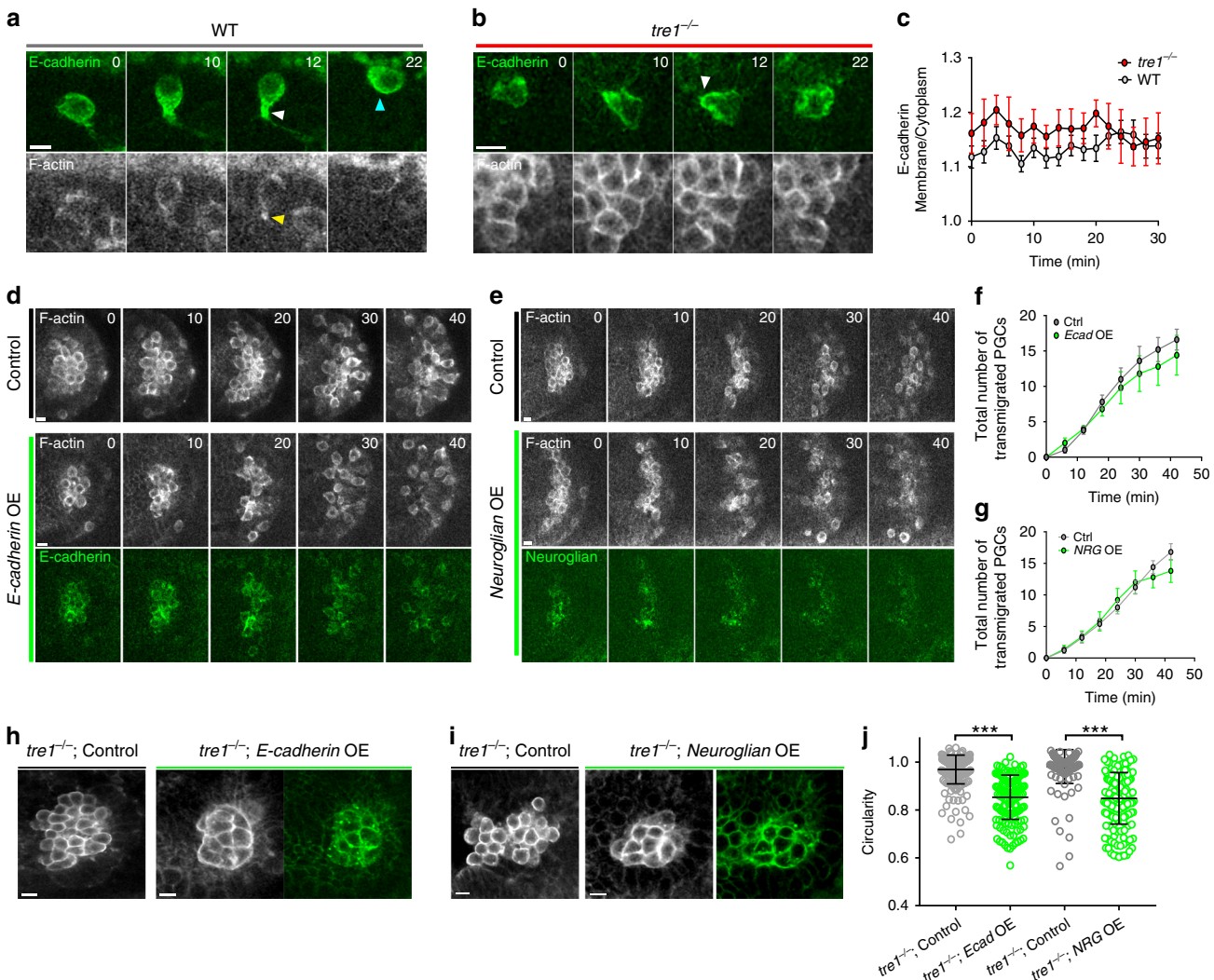

**Fig. 6 PGC disperse without altering E-cadherin levels and can overcome increased adhesion.** Two-photon timelapse imaging of representative transplanted WT (**a**) or *tre1⁻ᐟ⁻* (**b**) PGCs expressing E-cadherin-3xGFP and lifeact-tdTomato within lifeact-tdTomato expressing WT (**a**) or *tre1⁻ᐟ⁻* (**b**) embryos. White arrow marks region of E-cadherin enrichment and yellow arrow marks an F-actin focus. Cyan arrow marks posterior where E-cadherin remains enriched after separation is complete. Representative images are from n = 7 embryos in (**a**) and n = 5 embryos in (**b, c**) Quantification of E-cadherin membrane/cytoplasm ratio in transplanted WT (n = 16 PGCs from 7 embryos) and *tre1⁻ᐟ⁻* (n = 9 PGCs from 5 embryos). Error bars are SEM. Two-photon timelapse imaging of a representative control (top panels), E-cadherinmClover2 (**d**) overexpressing (bottom panel), or Neuroglian-mClover2 (**e**) overexpressing (bottom panel) PGC cluster for 40 min post endoderm EMT. Representative images in (**d**) and (**e**) are from n = 5 embryos in both control and experiment. **f–g** Quantification of the total number of PGCs which have transmigrated in control (n = 5 embryos) vs. E-cadherin-mClover2 overexpressing (n = 5 embryos) (**f**) or control (n = 5 embryos) vs. Neuroglian-mClover2 overexpressing (n = 5 embryos) (**g**) for 40 min after endoderm EMT. Error bars are SEM. Representative Z slice from *tre1⁻ᐟ⁻* control (left), *tre1⁻ᐟ⁻* Ecadherin mClover2 overexpressing (**h**), or *tre1⁻ᐟ⁻* Neuroglian mClover2 overexpressing (**i**) PGC cluster. Representative images in (**h**) are from n = 7 embryos in control and n = 8 embryos in experiment. Representative images in (**i**) are from n = 6 embryos in control and n = 9 embryos in experiment. **j** Quantification of the circularity of *tre1⁻ᐟ⁻* PGCs under the following conditions-control (n = 220 PGCs from 7 embryos) vs. Ecadherin-mClover2 overexpression (n = 166 PGCs from 8 embryos) and control (n = 155 PGCs from 6 embryos) vs. NeuroglianmClover2 overexpression (n = 179 PGCs from 9 embryos). Data are presented as mean values +/− SD. Statistical comparisons are from a Mann–Whitney test with p < 0.0001 in both comparisons. ***p < 0.001. All times are in minutes. All scale bars, 10 μm. Source data from (**c**, **f**, **g** and **j**) are provided as a Source Data file.

## Discussion

Our work here harnesses two-photon live imaging to provide an in vivo description of how actomyosin contractility is deployed to disperse cell clusters lacking apical–basal polarity under physiological conditions. In contrast to current models of epithelial delamination, cluster dispersal does not involve a sustained downregulation of cell–cell adhesion[9] or augmented force production[3,6,8] and is surprisingly robust to increased levels of adhesion (Fig. 6d–g). Rather, inherent migratory forces are co-opted to liberate cells (Fig. 8). This is

accomplished through the sensing of a directed migration cue via the GPCR, Tre1[12,20,22]. Tre1 signaling stabilizes and orients migratory polarity radially from the cell cluster, thereby positioning posterior myosin II dependent contractile forces towards cell–cell interfaces in the cluster interior. This collective radial polarity (Fig. 3e, h) stabilizes cell–cell interfaces and enables symmetric tugging, increasing the efficiency of cluster dispersal (Fig. 7). Symmetric tugging, however, is not absolutely necessary for cell separation, as individual WT PGCs can still detach from *tre1* PGC clusters, albeit less efficiently

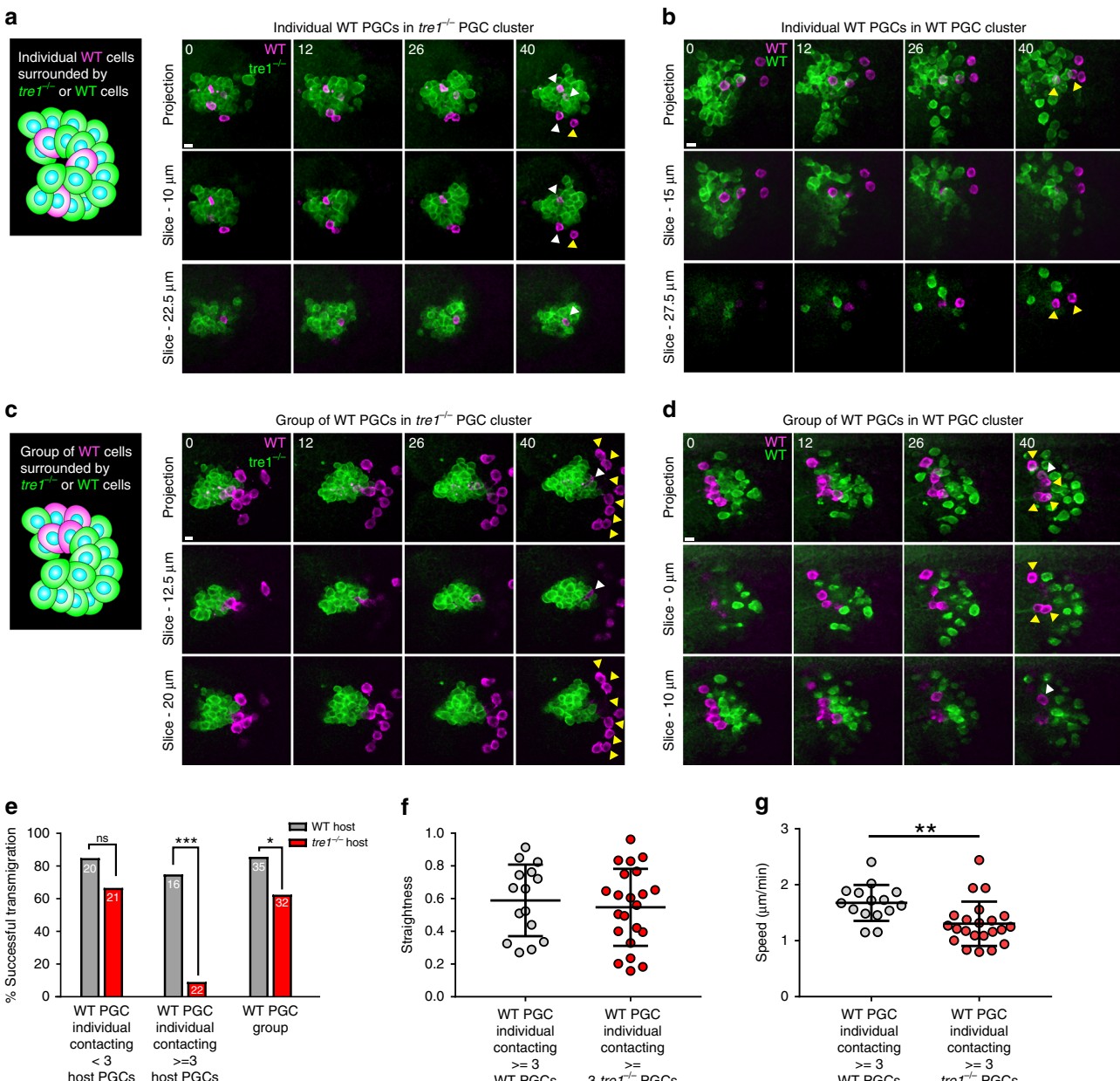

**Fig. 7 Coordination is necessary for efficient cluster dispersal.** Two-photon timelapse imaging of representative transplanted individual WT PGCs into *tre1*$^{-/-}$ (**a**) or WT (**b**) hosts or transplanted group of WT PGCs (Defined by 3 or more contacting PGCs) in *tre1*$^{-/-}$ (**c**) or WT (**d**) hosts. Transplanted PGCs are marked with lifeacttdTomato while host PGC clusters express Moesin-actin binding domain (ABD)-GFP. Top row of images is a maximum intensity projection while bottom two rows show a Z slice at the indicated depth from the top of the cluster. White arrows mark PGCs which have not detached from the cluster while yellow arrows mark PGCs which have detached and transmigrated. Representative images are from *n* = 22 embryos in (**a**, **c**) and *n* = 14 embryos in (**b**, **d**). Scale bars, 10 μm. (**e**) Quantification of successful transmigration events from transplanted individual WT PGCs (left and middle set of columns) and WT PGC groups (right set of columns). Different hosts are color coded. The number of PGCs in each group is indicated on the bars taken from *n* = 14 WT host embryos and *n* = 22 *tre1*$^{-/-}$ host embryos. Statistical comparisons are from a Fisher's exact test with *p* < 0.0001 in the central comparison and *p* = 0.0478 in the comparison on the right. ***p < 0.001 and *p <0.05. Quantification of straightness (**f**) and speed (**g**) of transplanted WT PGCs contacting > = 3 WT (*n* = 15 from 14 embryos) or *tre1*$^{-/-}$ PGCs (*n* = 22 PGCs from 22 embryos). Data are presented as mean values +/- SD. Statistical comparisons are from a Mann-Whitney test in (**g**) with *p* = 0.0021. **p < 0.01. Source Data from (**e**–**g**) is provided as a Source Data file.

(Fig. 7a, e). Subsequent detachment requires sustained pulling on cell–cell adhesions provided by a stable migratory polarity. Thus, randomly migrating cells, equally capable of contractile force production, are unable to separate because they do not pull on cell–cell adhesions in a given orientation for a sufficient period of time. A caveat to our model is that we have not directly shown that migrating PGCs exert posterior pulling forces, as this is technically challenging

at the depth where PGC cluster dispersal occurs. However, posterior pulling forces have been clearly demonstrated in various cell types utilizing a rearward driven 3D migration mode which closely resembles PGC migration in *Drosophila*[44,45].

Mechanistically, the migration-based cluster dispersal mechanism we outline here harbors many commonalities with hepatocyte growth factor (HGF) mediated epithelial scattering[2,46,47]. Myogenic

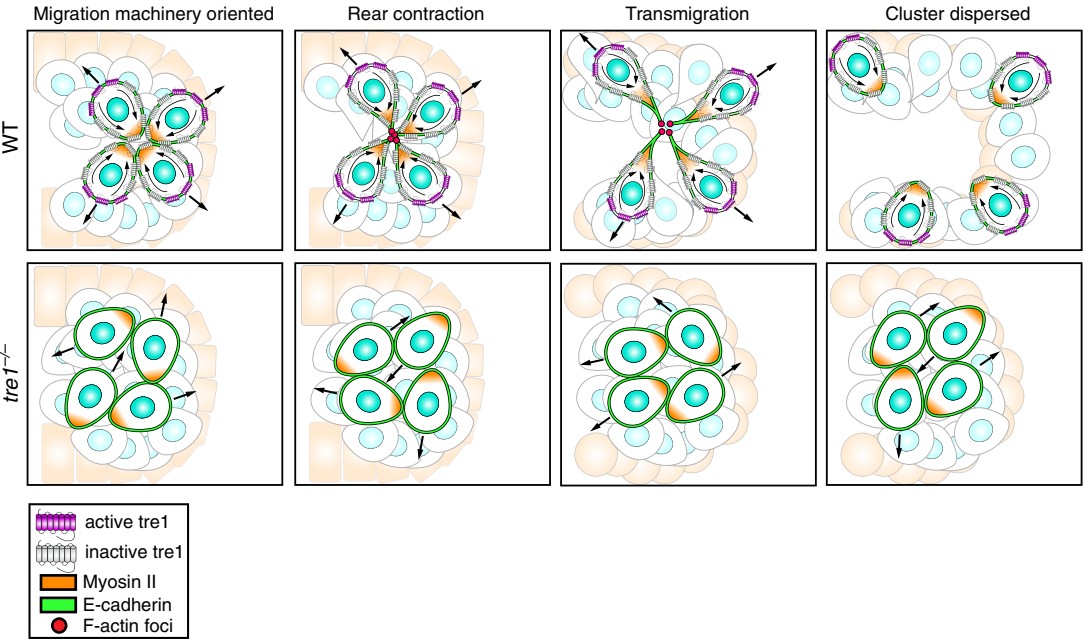

**Fig. 8 PGC clusters disperse by directing individual migratory polarity outward to collectively remove cell–cell adhesions.** PGC clusters disperse by utilizing Tre1−/− to interpret a presumed migratory cue presented at the exterior of the cluster to orient and stabilize individual migratory polarity radially from the cluster. Sustained polarization of myosin II contracts posterior cortices into foci as PGCs separate. Across the cluster, the opposing directions of PGC migration facilitate a collective response by aligning diametric migratory forces, akin to breaking a rope (cell adhesions) by pulling both ends apart. E-cadherin remains present on the cell membrane throughout. In the absence of Tre1−/−, PGC motility is randomized, resulting in a significant decrease in migration persistence. Clusters remain intact as PGCs are unable to orient their migration in a given direction for a sufficient time period to detach.

precursors induced to delaminate by ectopic application of HGF maintain expression of N-cadherin, the cardinal adhesion molecule originally linking them to the dermomyotome[46]. Similarly, HGF induced scattering of Madin–Darby canine kidney (MDCK) cells does not involve direct alterations in E-cadherin[47,48]. Instead, HGF promotes motility and strengthens cell-ECM attachment through integrins, which in turn generate a local increase in tension on cell–cell adhesions until they are physically disrupted. PGCs, on the other hand, continue to utilize E-cadherin to adhere to another cellular substrate, the surrounding endoderm, to pull away from each other. For both PGCs and HGF stimulated MDCK cells, the absence of free space to migrate is sufficient to block dispersal[21,48].

Cell ensembles frequently exhibit collective migration during development and disease[49]. How is group cohesion maintained if contractile migratory forces are sufficient to disrupt cell–cell adhesion? In collectively migrating squamous cell carcinoma (SSC) cells[50], primary colorectal cancer explants[51], and *Xenopus* neural crest[52], actomyosin contractility is enriched along the group perimeter and is actively suppressed from cell to cell interfaces in SSC cells to prevent cell detachment. This suppression relies on Discoidin domain receptor 1 (DDR1), which acts in a noncanonical manner at cell–cell interfaces by recruiting Par3 and Par6 to control the localization of RhoE to antagonize RhoA activity. Depletion of DDR1 and other members of the complex result in elevated levels of active Myosin II at cell–cell margins and individual cell migration away from the group, leading to group dispersal[50]. This is strikingly similar to the mechanism we have uncovered here during PGC cluster dispersal and that of HGF mediated cell scattering[47]. Recent work has also revealed an alternative strategy to reduce RhoA signaling at cell–cell junctions of collectively migrating SSC cells through Snail dependent expression of claudin-11, which activates Src to recruit p190RhoGAP to cell–cell interfaces[53]. In collectively migrating Drosophila border cells, E-cadherin tension is also reduced in the interior of the migrating cell cluster[54]. Thus, while the suppression of contractile forces at cell–cell contacts appears to

be a general principle to ensure cohesion in collectively migrating cell populations, PGCs actively direct actomyosin contractility towards cell–cell interfaces to separate. It will be interesting to assess whether pathological cell aggregates[55] can be coaxed to disperse through a similar mechanism.

Our work here demonstrates that cluster dispersal can be driven by the concerted reorientation of migratory actomyosin forces towards cell–cell interfaces. For PGCs, this is accomplished by sensing a directed migration cue through a GPCR, Tre1. However, in principle, any migratory cue, such as ECM or substrate stiffness, could be sufficient to mediate this reorientation. Subsequent cell–cell separation does not require any alterations of the inherent actomyosin forces driving migration. Rather, these forces must be applied on cell–cell junctions for a sufficient period of time, highlighting a potential safeguard against erroneous cell detachment. An open question is the identity of the Tre1 ligand. Given that PGCs are directed to migrate toward the endoderm within a tightly enclosed pocket, the ligand is likely to be surface bound, which would concur with the known roles of Tre1 in orienting neuroblast division[56] and immune cell extravasation[57]. Overall, we anticipate that directed motility-based separation will have general relevance for individual cell detachment events from clustered cell groups lacking apical–basal polarity in development and disease.

## Methods

**Fly strains**. All fly strains were reared at 25 °C and are listed in Supplementary Table 1. Transgenic lines were generated by Bestgene Inc. via phiC31 integrase-mediated transgenesis. Landing sites used were su(Hw)attP8 on the X chromosome, attP40 and su(HW)attP5 on the second chromosome, and attP2 and VK27 on the third chromosome.

**Constructs**. All cloning was performed using Infusion (Clontech) and all PCR was carried out using Q5® High Fidelity DNA polymerase (NEB). All primer sequences are available in Supplementary Table 2. pWALIUM22[58] was used as the backbone for UAS and *nanos (nos)* driven constructs. For *nos* driven constructs, UAS sites

and the K10 3′UTR were removed from pWALIUM22 via PCR and replaced with *nos* regulatory elements[59] (*nos* promoter, 5′UTR and *nos* 3′UTR). The *nos* 5′UTR and 3′UTR were separated by a ZraI restriction site. ORFs were amplified by PCR and inserted into this plasmid following digestion with ZraI. For UAS driven constructs with the *nos*TCE-*pgc* 3′UTR, the K10 3′UTR was removed from pWALIUM22 via PCR and a spacer DNA sequence (Gprk2 ORF (DGRC-LD21923)), a ZraI restriction site, and the first 100 bp of the *nos* 3′UTR coupled to the *pgc* 3′UTR[59] were placed in. ORFs were amplified via PCR and inserted into this plasmid following digestion with NheI and ZraI, which released the spacer. Construction of the RhoA sensor has been described previously[60].

*Nos-Lifeact-tdTomato-P2A-tdKatushka2-CAAX (Lifeact-tdTomato).* *Nos* regulator elements drive Lifeact[61] fused to tdTomato (Addgene 54653), a P2A peptide, and tdKatushka2 (Addgene 56041) with a CAAX box from human KRAS. Two fragments were amplified via PCR and cloned into pWALIUM22 with *nos* regulatory elements described above—(1) Lifeact sequence was appended to tdTomato via a primer and (2) P2A and CAAX sequences were added to tdKatushka via primer. Fly lines were generated on attP40 and su(HW)attP5 on the second chromosome and attP2 and VK27 on the third chromosome.

*UASp-degradFP-nosTCE-pgc 3′UTR.* The degradFP system (Addgene- 35579) was amplified via PCR and inserted into pWALIUM22 containing the *nos*TCE-*pgc* 3′UTR described above. Fly lines were generated on su(Hw)attP8, attP40, and attP2.

*UASp-DE-cadherin-mClover2-nosTCE-pgc 3′UTR.* The *DE-cadherin* ORF was amplified via PCR from pUAST-DEFL[62] with a linker and introduced along with a second fragment, mClover2 (Addgene- 54577) into pWALIUM22 containing the *nos*TCE-*pgc* 3′UTR described above. Fly lines were generated on attp40 and attP2.

*UASp-Neuroglian-mClover2-nosTCE-pgc 3′UTR.* The *Neuroglian* ORF (DGRC-GH03573) was amplified via PCR with a linker and introduced along with mClover2 (Addgene- 54577) into pWALIUM22 containing the *nos*TCE-*pgc* 3′UTR described above. Fly lines were generated on attp40 and attP2.

*UASp-mClover2-Rho1-G14V-nos 3′UTR, UASp-mClover2-Rho1-G14V-gcl 3′UTR, UASp-mClover2-Rho1-G14V-pgc 3′UTR, UASp-mClover2-Rho1-G14V-nosTCE-pgc 3′UTR.* The G14V mutation was introduced via site directed mutagenesis (NEB, Q5® site directed mutagenesis kit-E0554S) of Rho1 (DGRC-LD03419). The Rho1-G14V ORF was amplified via PCR and introduced along with mClover2 (Addgene-54577) into pWALIUM22. Different 3′UTRs were amplified via PCR from previously described constructs[59] and used as fragments in infusion cloning reactions with mClover2-Rho1-G14V.

*MatTub-Gal4.* A weaker maternal driver which utilizes Gal4 instead of Gal4-VP16 was generated to reduce toxicity from overexpression of UAS constructs. Three fragments were cloned into pWALIUM22 with UAS sites and K10 3′UTR removed—(1) αTub67C promoter and 5′UTR were amplified via PCR from genomic DNA from P{matα4-GAL-VP16}67; P{matα4-GAL-VP16}15 females (BL 80361), (2) Gal4 ORF was amplified from pBPGAL4.2Uw-2 (Addgene-26227), and (3) αTub84B 3′UTR was ordered as a Gblock from IDT technologies. Fly lines were generated on attP40 and attP2.

**Live imaging.** All embryos were produced at 25 °C. To prepare for live imaging, embryos were first dechorionated in 50% bleach for 3 min, extensively washed, and collected onto apple juice agar plates. After visual staging, embryos were oriented with their dorsal surface facing up, glued onto #1.5 glass coverslips (ThermoFisher Scientific, 12-544-BP) with heptane glue, covered with halocarbon oil 27 (Sigma, H8733), and placed onto a gas permeable membrane (YSI, 098094). Live imaging was performed on a Prairie Ultima (Bruker technologies) with a Zeiss C-Apochromat 63x, 1.2 NA water objective utilizing a 4W Ti:Sapphire Chameleon Ultra I and Chameleon Ultra II laser (Coherent) driven by PrairieView 4.3.2.24. Images from Supplementary Fig. 2c–e were taken with a Nikon CFI Apo IR 60 × 1.27 NA water objective with a Chameleon Discovery with total power control.

A custom filter cube consisting of bandpass filters—ET575/50m-2p and ET660/60M-2p and a T612LPXR-UF1 dichroic was used to simultaneously visualize tdTomato and tdKatushka2 under 1050 nm excitation. Images in Supplementary Fig. 2c–e were taken with 1080 nm excitation. A filter cube consisting of bandpass filters—ET510/50m-2p and ET575/50m-2p and a T540lpxr dichroic was utilized to simultaneously visualize GFP and tdTomato. For most experiments, 980 nm was utilized to image GFP and tdTomato. Longer wavelengths were utilized when imaging stronger GFP lines to favor excitation of tdTomato, while shorter wavelengths were used with weaker GFP lines. All filters were purchased from Chroma Technology Corp.

To visualize the distribution of GFP-tagged binding domains and proteins (DiaRBD-GFP, GFP-ROCK, Pak1-RBD-GFP, and Wasp-RBD-GFP), PGC clusters were first identified with the PGC restricted tdTomato-Anillin-RBD-P2A-tdKatushka2-CAAX under 1050 nm excitation. Two consecutive Z stacks were taken—the first was at 1050 nm to simultaneously image tdtomato-Anillin-RBD and the membrane marker, tdKatushka2-CAAX. The filter cube was quickly exchanged, the laser wavelength was switched to 980 nm, and a second Z stack of the GFP-tagged constructs and tdtomato-Anillin-RBD was obtained. Most of the resulting drift between Z-stack acquisitions in embryos occurred in Z (~2 μm drift) and the stacks were registered through comparison of tdtomato-Anillin-RBD, which was visualized in both stacks.

**Image processing and analysis.** Images presented in figures and movies were denoised using the CANDLE[63] package for Matlab in Matlab 2016a (Mathworks). The denoising settings utilized were beta = 0.3, patch radius = 2, and search radius = 2. All quantitative analysis was performed on raw data except for cell tracking. Cell tracking in Fig. 1 was done in Imaris 8.0.2 (Bitplane Inc.), while tracking in Figs. 4 and 7 was performed in ImageJ and Matlab. Nuclei (H2B-GFP) and lifeact-tdTomato images were imported into Imaris and the "Spots" function was used to identify nuclei across all frames. PGCs were identified by expression of lifeact-tdTomato and larger nuclear size. Tracks were automatically generated, manually corrected, and final XYZ positions were imported into Matlab for analysis. For cell tracking in ImageJ and Matlab, regions of interest (ROI) were first defined in ImageJ for cell segmentation. ROIs were subsequently imported into Matlab and cells were automatically segmented from ROIs using the "func_threshold" function[64] from all Z planes. The cell centroid was calculated as the average XYZ pixel value for all segmented pixels. All other analysis was completed using a combination of ImageJ and custom scripts in Matlab described below.

To quantify posterior area and lifeact-tdTomato intensity, a ROI encompassing the posterior PGC cortex up to the nucleus was selected in a single Z slice harboring the greatest area of the PGC. The posterior area was defined as the number of pixels in the ROI and the lifeact intensity was the mean intensity in the ROI. Only PGCs which could be accurately tracked in Z were used in analysis.

To plot fluorescence intensity as a function of distance from cluster centroid, individual PGC cluster Z stacks were first manually segmented in ImageJ through user defined ROI, utilizing the PGC localized membrane marker (tdKatushka2-CAAX) or F-actin marker(Lifeact-tdTomato) as a reference. A second ROI was also defined in the background to account for intensity variations resulting from different depths. The ROIs were then imported into Matlab and used as masks for the other fluorescent channels. Next, the cluster centroid, defined as the average of all XYZ coordinates of the segmented pixels defining the cluster, was calculated. The intensity of each segmented pixel was normalized to the average of the background ROI at the same depth and its relative distance to the cluster centroid was computed. Lastly, the distance from the cluster centroid was discretized into 50 equally spaced bins and the average intensity of the pixels within the bins was plotted. For time dependent measurements of fluorescence intensity as a function of distance from centroid, the same operation described above was executed at each time point and the resulting data was interpolated using the "interp2" function in Matlab. The final data was visualized using the "pcolor" function in Matlab.

To analyze the fluorescence intensity of polarized myosin II in individual cells, regions of high myosin II, defined by being ≥1.2-fold higher intensity than cytoplasm, were manually segmented in ImageJ across all Z planes along with a corresponding region in the same cell in the cytoplasm. The ROIs were imported into Matlab and the ratio between the ROI defining high myosin II and the ROI in the cytoplasm was calculated as polarized myosin II intensity (See schematic in Fig. 3c). To analyze the relative orientation of polarized myosin II with respect to the endoderm center, the endoderm center was first manually defined based on endoderm boundaries in myosin II fluorescent images. On an individual cell basis, the angle between vectors from the PGC nucleus to the region of polarized myosin II and the PGC nucleus to the endoderm center was used to calculate myosin II orientation (see schematic in Fig. 3d).

To quantify posterior myosin II intensity over time, individual PGCs were first segmented in ImageJ by manually defining ROIs around cell boundaries in the Z plane which contained the greatest polarized myosin II intensity. Only PGCs which stayed within three Z planes over the time course of the experiment were used in analysis because the majority of the PGC cell body remained within a single Z plane. ROIs were subsequently imported into Matlab for segmentation and segmented PGCs were computationally rotated vertically with posterior towards the bottom so that the length of the cell body could be assessed by the number of segmented rows. The posterior was defined by the lowest 20% of segmented rows and mean intensity of myosin II in this region was calculated as the myosin II RLC posterior intensity.

To calculate myosin II orientation in individual PGCs over time, maximum intensity projections were used in analysis and individual PGCs were first manually segmented in ImageJ using coarse ROIs. ROIs were imported into Matlab and segmentations were refined using the "func_threshold" function described above. Regions of polarized myosin II were defined by first isolating the periphery of individual PGCs by eroding the PGC segmentation and subtracting the eroded image, followed by identifying regions >10 pixels which were >1.5 standard deviations above the mean intensity. Orientation was then calculated by the angle between vectors from the PGC centroid (calculated from segmentation) and the polarized myosin II centroid and from the PGC centroid to the endoderm centroid (manually defined based on endoderm boundary from lifeact-tdTomato).

To calculate the membrane to cytoplasm ratio of E-cadherin-3xGFP, PGCs were first manually segmented in ImageJ in the GFP channel using the Z plane which contained the greatest cell area and the segmentations were imported into

Matlab. The PGC membrane signal was isolated by eroding the PGC segmentation and subtracting it from the original segmentation. The eroded image was used as the cytoplasm signal.

To calculate circularity, individual PGCs were manually segmented in ImageJ using the Z plane with the greatest cell area and imported into Matlab. The segmented PGC area and perimeter were calculated using the "regionprops" command and subsequently used in the following formula, where $A$ = area and $P$ = perimeter.

$$\text{Circularity} = 4\pi A/P^2. \tag{1}$$

**Cell culture, transfection, and imaging**. S2 cells were maintained in Schneider's medium (ThermoFisher, 21720-001) supplemented with 10% (v/v) fetal bovine serum (ThermoFisher, 16140071) and 1% (v/v) penicillin streptomycin (Thermo-Fisher, 15140122). Constructs expressed in S2 cells were driven by the act5c promoter by subcloning into Ac5-Stable2-neo (Addgene #32426). Transfections were carried out using Effectene (Qiagen, 301425) following manufacturers recommendations. Transfected S2 cells were plated onto Labtek slides (ThermoFisher, 155409) coated with a 50 μg/ml solution of Concanavalin A diluted in PBS (Cayman Chemical, 14951) for 1 h at room temperature. S2 cells were subsequently imaged on a widefield Nikon Eclipse Ti using a Plan-Apochromat 60 × /1.4 NA oil objective.

**Immunofluorescence**. Appropriately aged embryos were first dechorionated in 50% bleach for 3 min, washed, collected on a nylon mesh, and placed in a scintillation vial containing a 1:1 (v/v) mixture of heptane and 4% paraformaldehyde (Electron Microscopy Sciences, 15714-S) in PBS on a shaker for 20 min. The paraformaldehyde was subsequently removed and replaced with methanol and the scintillation vial was vigorously agitated by hand for 30 s to remove the vitelline membrane. Embryos were then stored in methanol at −20 °C until subsequent processing. Embryos stored in methanol were gradually rehydrated with PBST (0.3% Triton X-100 (Sigma, T8787)) and blocked in PBST with 1% bovine serum albumin (BSA) (Sigma, A4503) for 30 min at room temperature. Primary antibodies utilized in this study—rabbit anti-Rab5 (1:500, Abcam, ab31261), mouse anti-Rab7 (1:100, Developmental Studies Hybridoma Bank), chicken anti-GFP (1:500, Aves, GFP-1010), rabbit anti-vasa (1:5000, R. Lehmann), and goat anti-vasa (1:500, Santa Cruz Biotechnologies, sc-26877) were diluted in PBST with 1% BSA and applied overnight at 4 °C. After extensive washing, appropriate secondary antibodies (1:500, Jackson Immunoresearch) were diluted in PBST with 1% BSA and added to samples for 3 h at room temperature. Embryos were washed and subsequently mounted in Vectashield (Vector Laboratories, H-1000) and imaged with a Zeiss LSM 800 using Zen Blue 2.3 with a 40 × 1.3NA oil objective using a pinhole size of 1 AU.

Ovaries were dissected from 2- to 3-day-old females and were fixed in a 5% formaldehyde solution diluted in PBS for 25 min and subsequently incubated with PBST (1% Triton X-100) for 1 h at room temperature. Ovaries were then incubated with PBSTB (0.1%Triton X-100, 1% BSA) for 1 h and primary antibodies (described above)—rabbit anti-Vasa (1:3000) and chicken anti-GFP (1:500) were applied to samples overnight at 4 °C. On the next day after three 10 min washes in PBSTB, appropriate secondary antibodies diluted 1:500 in PBSTB were incubated with samples for 2 h at room temperature. Alexa Fluor 488® Phalloidin (ThermoFisher, A12379) diluted 1:500 in PBSTB was added in this step. After three 10 min washes in PBSTB, ovaries were mounted with Vectashield and imaged on a Zeiss LSM 800 using Zen Blue 2.3 with a Plan-Apochromat 20×/.8 NA air objective using a pinhole size of 1 AU. Z slices through the entire depth of ovarioles were taken with a spacing of 2 μm.

**Development of the nosTCE-pgc 3′UTR**. Current techniques for conditional expression in PGCs rely upon maternal deposition of Gal4-Vp16 at the posterior using nos-Gal4-VP16 and subsequent zygotic transgene expression beginning at stage 9 of embryogenesis through a paternally contributed UAS transgene. Transgene expression thus begins during PGC cluster dispersal and may not be expressed at sufficient levels to affect dispersal. To circumvent this issue, our goal was to identify an appropriate 3′UTR which would allow conditional maternal transgene expression in PGCs. To identify 3′UTRs which could allow conditional maternal expression in PGCs without affecting oogenesis, we generated UASp test constructs driving mClover2-Rho1-constitutively active (CA, G14V mutation) with the 3′UTRs of known posterior localized mRNAs, nos, gcl, and pgc (Supplementary Fig. 5). These flies were crossed to females harboring a maternal driver, MatTub-Gal4 (described above), and the F1 progeny were analyzed. We reasoned that undesirable translation of Rho1-CA during oogenesis would be evident by defective egg production. In the ovaries of females expressing mClover2-Rho1-CA nos 3′UTR and mClover2-Rho1-CA gcl 3′UTR, mClover2-Rho1-CA expression was detected during oogenesis and lead to multinucleated nurse cells and aberrant nurse cell morphology (Supplementary Fig. 5a, b). These females did not produce any eggs. In contrast, in the ovaries of females expressing mClover2-Rho1-CA pgc 3′UTR, mClover2-Rho1-CA was not expressed during oogenesis and nurse cells had a WT morphology. These females produced eggs (Supplementary Fig. 5c). Examination of stage 5 embryos from MatTub > mClover2-Rho1-CA pgc 3′UTR females revealed that mClover2-Rho1-CA was expressed essentially uniformly

across the embryo (Supplementary Fig. 5d), leading to abnormal PGC morphology and cellularization defects. To decrease somatic expression of mClover2-Rho1-CA, we reasoned that the addition of the nos translation control element (TCE)[65], which prevents Nos translation in the bulk cytoplasm, to the pgc 3′UTR would similarly decrease the translation of unlocalized mRNA. Thus, we generated a UASp-mClover2-Rho1-CA nosTCE-pgc 3′UTR line and followed the crossing scheme described above. Strikingly, ~36% of stage 5 embryos from MatTub > mClover2-Rho1-CA nosTCE-pgc 3′UTR females (4/11) were devoid of PGCs and exhibited WT cellularization (Supplementary Fig. 5e), which was never observed in the embryos of MatTub > mClover2-Rho1-CA pgc 3′UTR females. These results imply that when high levels of mClover2-Rho1-CA nosTCE-pgc 3′UTR mRNA are deposited into embryos, local translation of Rho1-CA prevents PGC formation and reduced translation of unlocalized mRNA allows cellularization to proceed. We speculate that both processes fail when similar levels of mClover2-Rho1-CA pgc 3′ UTR mRNA are deposited into embryos. In embryos with moderate levels of mClover2-Rho1-CA expression, as evidenced by successful PGC formation and the initiation of gastrulation, the nosTCE-pgc 3′UTR enabled a greater enrichment of mClover2-Rho1-CA in PGCs relative to surrounding cells than the pgc 3′UTR alone (Supplementary Fig. 5d, e).

**Optogenetics**. Optogenetic experiments were carried out using a Chameleon Ultra II laser for imaging and a Chameleon Ultra I laser for photo-activation. The bipartite opto-RhoGEF system[36], consisting of UASp-mCherry-RhoGEF2-CRY2 and UASp-CIBN::pmGFP, was zygotically expressed solely in PGCs using Nos-Gal4-VP16 females. mCherry-Rhogef2-CRY2 required very strong excitation for detection which was not amenable for live imaging; therefore, we utilized CIBN::pmGFP to track PGCs. PGCs migrating in the lateral mesoderm were first identified by imaging CIBN::pmGFP using 1010 nm excitation to avoid activation of the system. Subsequently, a timelapse experiment was executed, with three Z slices spaced by 2 μm imaged every 30 s under 1010 nm illumination to track PGCs. An ROI (~5 μm by 10 μm) was defined at the leading edge of a PGC (facing the anterior of the embryo) and illuminated with 950 nm light. PGC movement following photo-activation was defined as a reversal if the PGC migrated out of the ROI.

**Statistics**. All experiments were performed with at least two independent replicates. All statistical comparisons were carried out in Prism (GraphPad). Two sided Mann–Whitney tests were used for pair-wise group comparisons, while Fisher's exact test was used to compare proportions.

**Reporting summary**. Further information on research design is available in the Nature Research Reporting Summary linked to this article.

## Data availability

All data is available from the corresponding authors upon reasonable request. Raw data from Figs. 1–7 and Supplementary Figs. 2–4 are provided as a Source Data file. Source data are provided with this paper.

## Code availability

Matlab scripts utilized in this study are available at—https://github.com/linb06/PGC-dispersal.

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

## Acknowledgements

We thank X. Chen, Y. Bellaïche, S. De Renzis, and the Bloomington *Drosophila* Stock Center for fly stocks; S. Hayashi, Addgene, and the *Drosophila* Genomics Resource Center for reagents; N. Yamaguchi, L. Barton, T. Lieber, T. Banisch, H. Knaut, J Nance, T. Inoue, and A. Ewald for comments on the paper; and members of the Lehmann lab for discussion.

This work is supported by NIH grant R37HD41900 to R.L. B.L. is a New York Stem Cell Foundation Druckenmiller Fellow. R.L. is a Howard Hughes Medical Institute investigator.

## Author contributions

B.L. and R.L. conceived the project. B.L. and J.L. performed experiments. B.L. and R.L. wrote the paper with input from all authors.

## Competing interests

The authors declare no competing interests.
