## [Peer Review File · Nature Communications]

Reviewers' comments:

Reviewer #1 (Remarks to the Author):

Collective cell migration plays a critical role in many events of normal development and homeostasis, and, it is becoming increasingly clear, is also a major mode of migration of metastatic cancer cells. Often, these migratory events involve cells with epithelial apical-basal polarity, but there are many examples in which cells disperse from non-epithelial clusters. The authors have developed *Drosophila* germ cell migration as one of the premier models for these processes. Here they address a key question: how do cells disperse from clusters lacking apical-basal polarity, a hallmark of advanced epithelial cancers? They use exceptionally powerful (and complicated!) genetic, cell biological and image analysis tools to dissect this. They demonstrate that germ cells exhibit radial migratory and parallel actomyosin polarity, that this requires the guidance receptor *Tre1*, that polarized myosin activation is critical, and that dispersal does NOT require reduced levels of E-cadherin adhesion. Their data are lovely, their analysis compelling and their conclusions very well supported. This is an exceptionally lovely study that will be widely read by cell and developmental biologists and I strongly support publication.

Minor issues

Abstract. I found the description that cells orient “posterior migratory forces toward the cluster interior” confusing, as the direction of migration is radially away from the cluster center.

Fig. 1A,H. Add information about when the endoderm EMT occurred.

I'd love to see Fig S1/S2 included in the main Figures—this is among the most impressive uses of the Rho sensor I have ever seen. Is Rho itself polarized?

Figure 3. Do individual transplanted wildtype cells prematurely penetrate the endoderm?

p. 10, line 273. I might suggest slightly toning down the “sufficient” conclusion. They can clearly reverse migration with Rho activation in a somewhat artificial way but saying this demonstrates sufficiency is a bit much.

Reviewer #2 (Remarks to the Author):

Summary:

This manuscript by Lin et al. focuses on the mechanism of migration of primordial germ cells (PGCs). PGCs form originally a cluster from which cells individualize and migrate away in opposite directions. In this manuscript, the group of Ruth Lehmann revisits their own previous work that involved the receptor *Tre1* in the polarization of Rho activity. They show that local contractility induced by Rho and Myosin II at the back of the PGCs directs their migration. Elegant optogenetics experiments show that local activation of Rho and hence Myosin II directs migration (although not extremely

efficiently). In addition, they show that the detachment and migration of PGCs does not require the downregulation of adhesion (DE-cadherin, neuroglian), and propose that other Rho GTPases (Rac and Cdc42) are not involved in PGCs migration. Finally, they show that contractility needs to be collectively polarized, which seems to be the only true new findings. New imaging methods and analysis give an exquisitely detailed description of the detachment and migration of PGCs away from their cluster. However, most of the observations are not entirely novel and despite a clever discussion, the main message of the article seems redundant to previous work which strongly reduced my enthusiasm.

Major concerns:

- The first major conclusion of the manuscript is that forces formed at the back of PGCs direct their migration out of the cluster. This is inferred from the localization of contractility in the actomyosin cytoskeleton (observed by different markers: pMLC2 and tagged Myosin II) and by PGCs morphological changes. There is no direct measurement of forces, which is obviously hard if not entirely impossible in this in vivo system. Still, the description should be more precise, I understand that it is extremely likely that contractility and morphological changes are reporter of forces, but it is not clear in which direction the forces is generated although the observed migration correlates with the generation of propulsive forces at the back.

- Optogenetics experiments show that Rho activation can modify the directionality of PGCs' migration. That said, the optogenetic results are not very compelling. Indeed, the effect is weak as less than a third of the PGCs change directionality when Rho is locally activated. Indeed 72%, i.e. a vast majority did not. The author should at least explain why the majority does not change direction. An important control, would also be to show that local activation of Rac (photoactivable Rac is available in Drosophila) is not able to change directionality (as the migration of PGCs was shown to be independent of Rac activation by the Lehmann group (Kunwar et al., PLoS Biology 2003)). (Also: there is no statistical analysis on Fig.4g).

- The second major conclusion is that Tre1 is mediating this contractility. This is not really new as previous work from the same group showed that Tre1 acts through Rho1 (Kunwar et al., PLoS Biology 2003) and that Tre1 polarizes Rho1 activity (Kunwar et al., JCB 2008; Leblanc and Lehmann, JCB 2017)). Since Myosin II is directly activated downstream of Rho1, the result is not surprising although it is obviously important to demonstrate that the canonical Rho1 pathway is involved.

- Again, the conclusions about adhesion are not entirely novel as the Lehmann group already showed that the modulation of DE-cadherin was not impacting on migration (Kunwar et al., JCB 2008). The reasons to test neuroglian are not very compelling.

- The data about the requirement for a collective movement of PGCs in opposite direction is interesting, but I am wondering if there can be other explanations (for example that when WT PGCs adhere to multiple Tre1^{-/-}, they cannot migrate properly as they have a charge to move). I such I am not entirely convinced that it is the collectiveness of movement that is important and I don't know how to test this in a more compelling way.

- The use of the word "sensor" for the Rho, Rac or Cdc42 activity when using a simple binding

domain is, in my opinion, unproper. Indeed, FRET sensors working in vivo in *Drosophila* for Rac and Cdc42 exist (I don't think that there is a Rho sensor). These are indeed activity sensor as they rely on the proportion of active GTPase, although they also have some caveats. Binding domains are recruited by active Rho GTPase, but they display the localization of active GTPase properly only if the proportion of binding domain is in the range of the number of active Rho GTPase. If the binding domain is overabundant, it will be localized at regions that are not associated with active GTPase, and can even bind possibly to other GTPases. I know that most of the probes used here were well characterized by the group of Susan Parkhurst but their efficiency might dramatically differ from one cell type to another. As such, I think that the data of the anillin binding domain are convincing but the Rac and Cdc42 binding domains seem largely overexpressed which may impair the observation of the site of their active target GTPase.

- Also, again, previous work from the Lehmann group already showed that Rac and Cdc42 are dispensable for PGC migration (Kunwar et al., PLoS Biol 2003).

Reviewer #3 (Remarks to the Author):

This manuscript by Lin and colleagues investigates the mechanism for how initially cohesive groups of cells are able to ultimately disperse. The model used here is the *Drosophila* embryonic primordial germ cells (PGCs), which start out as a clustered rosette of cells but then separate via EMT and undergo transmigration through the endoderm. Previous work showed that the GPCR Tre1 is required for dispersal. However, the overall mechanism for how PGCs are able to separate from the cluster prior to migration through the endoderm is unclear.

The authors characterize the process of PGC dispersal using genetics, live cell imaging, biosensors, elegant transplantation experiments, and optogenetic tools. Rather than having a typical apical-basal polarity, PGC clusters have a front-rear polarity, with the rear (posterior) oriented to the center of the cluster. The posterior of individual wild-type (WT) PGCs is enriched with F-actin, suggesting polarized contractility. F-actin at the posterior continues to contract into bright foci as the PGCs disperse and migrate away. This polarized F-actin is lost in *tre1*^{-/-} mutant PGCs, which do not separate. Next, the authors use a RhoA GTPase biosensor, along with known RhoA effectors Dia and Rock, to show that RhoA-dependent contractility is similarly polarized to the rear of the PGCs. The activity of two other small GTPases (Rac and Cdc42), which in other cells can induce front-rear polarity, is not polarized. RhoA is known to activate myosin-II contractility in cells. Myosin-II (visualized by RLC-GFP) is enriched at the posterior of PGCs. Interestingly, myosin-II is stable in WT PGCs, but is less stable in *tre1*^{-/-} mutant PGCs. The authors transplanted WT or *tre1*^{-/-} mutant myosin-II-GFP-labeled PGCs into embryos lacking PGCs. This allowed better analyses of polarization of myosin-II with respect to PGC movement. Posterior-polarized myosin-II was retained in WT cells, but was re-oriented in *tre1*^{-/-} mutant PGCs. Downregulation of myosin-II in PGCs decreased the rate of PGC separation. Moreover, photoactivation of RhoA (and thus contractility) induced re-orientation of PGCs. Thus, the authors conclude that myosin-II-dependent contractile force is required for dispersal of PGCs. This dispersal does not require downregulation of cell-cell adhesion, as the PGCs retain E-cadherin. Additionally, raising the levels of either E-cadherin or another

adhesion protein, Neuroglian, did not prevent separation of PGCs. Finally, coordination of multiple PGCs is needed for dispersal of cells. This was shown by transplanting a few vs. larger groups of WT PGCs into *tre1*^{-/-} mutant embryos. The individual WT PGCs had difficulty undergoing transmigration, but larger groups of WT cells were mostly successful. The authors conclude that radial collective polarity through myosin-II-dependent contractility stabilizes cell-cell interfaces and enables symmetric tugging, and more efficient cluster dispersal.

This is a novel study using a number of cutting-edge tools to provide insights into the mechanisms of collective cell dispersal during development. There are implications for collective metastasis in cancer, as well as other migratory collectives during development. This manuscript will be of wide interest. The experiments appear to be rigorous, well-controlled, and are overall convincing. I have a few specific comments that would help clarify the results and discussion.

Comments:

1. This statement (lines 107-108) was confusing: "Subsequently, the foci were either tugged off of other PGCs and incorporated into trailing tails or were snapped off and left behind in the endoderm cavity (Fig. 1a)." I am unsure what the authors are referring to and this is not clear in the figure or in the movies. Additional labels on the figure, or panels showing this more clearly, may be needed.
2. The RhoA biosensor was validated in S2 cells, but not in PGCs (Fig. S1). It would be more convincing if the authors had co-expressed RhoN19 (DN) and RhoV12 (CA) with the biosensor in PGCs (if possible), to show similar relocation of the sensor to the cytoplasm with DN Rho and to the membrane with CA Rho. It is reassuring that two effectors of Rho (Dia and Rock) show similar localization with the RhoA biosensor (Fig. S2).
3. In Figure 2e-h, the authors show that myosin-II-GFP is polarized in the PGC cluster in WT but not *tre1*^{-/-} mutants. This is convincing, and is followed up in Figure 3 with the transplantation experiments. However, it seems that myosin-II-GFP is highly dynamic in *tre1*^{-/-} mutant embryos (Fig. 2f and Movie S6), suggesting that the contractility is unstable and thus PGCs cannot contract and disperse. I am unsure that Fig. 2h really captures the quantification of this dynamic myosin-II-enrichment; could the authors perform kymographs on their existing movies, across the PGC cluster in *tre1*^{-/-} mutant embryos to show fluctuations in GFP enrichment over time? Or is this just not possible because of the 3D nature of 30 cells in the embryo?
4. In the last section of the Results (lines 334-337; Fig. 6c-e), the wording is a bit confusing and could be clarified.
5. The model in Fig. 7 only focuses on individual WT or *tre1*^{-/-} mutant PGCs, and lacks a figure legend. If the point is for group migration and contraction to help disperse the cluster of PGCs, along with individual PGC polarity, it would be helpful to see that in the summary panels.

Minor comments:

1. In Fig. 1a-b, there are asterisks on a few cells. I assume these are the cells shown in the movies, but this could be mentioned in the figure legend.
2. As far as I can tell, Movie S7 is missing a call-out in the text. I assume this movie should be mentioned around lines 200-202.
3. The statistics for Fig. 4g (OptoRhoGEF migration reversal) are missing on the panel.
4. Fig. 6e, the y-axis is labeled "% Successful transmigration" but the numbers are from 0-1.0 and not 0-100%. I assume this is mislabeled.

Dear reviewers,

We would like to thank you for the positive feedback on our manuscript. Your insightful comments have helped us improve the quality and clarity of the manuscript. As outlined in detail below, we have performed additional experiments as recommended by reviewers, modified figures, clarified wording within the main text, and added additional text to more clearly state the conceptual advances of this work with regards to our previous findings.

In summary we consider the advance our manuscript as follows:

1. We demonstrate that inherent migratory forces rather than changes in adhesive properties can be co-opted to liberate cells. We find that Tre1 GPCR signaling stabilizes and orients migratory polarity radially from the cell cluster, thereby positioning posterior myosin II dependent contractile forces towards cell-cell interfaces in the cluster interior. This collective radial polarity stabilizes cell-cell interfaces and enables symmetric tugging, increasing the efficiency of cluster dispersal.
2. We demonstrate that components of contractile force generation (RhoA, MyoII) are polarized in Tre1 defective cells and not evenly distributed as we previously reported from studies based on observing fixed materials. This new result led us to propose a new model for cell dispersal. In this model PGC detachment requires sustained pulling on cell-cell adhesions, which is provided by a stable migratory polarity. Randomly migrating Tre1 defective cells, equally capable of contractile force production, are unable to separate because they do not pull on cell-cell adhesions in a given orientation for a sufficient period of time.
3. We find that cluster dispersal does not involve a sustained downregulation of cell-cell adhesion. While we showed previously that reduction in E-cadherin levels reduced the clumping of *tre1* mutant cells, experiments in our new manuscript track endogenous E-cadherin in the wild type and address the consequences of increasing adhesion. We find that increasing adhesion does not modify germ cell dispersal and endoderm transgression. Thus, changes in the cell adhesion program are not determinative for cell dispersal.

Below is a point by point breakdown of how we have addressed each concern. Author comments are in blue below and in the main text all text changes are highlighted in green.

Reviewer 1

Their data are lovely, their analysis compelling and their conclusions very well supported. This is an exceptionally lovely study that will be widely read by cell and developmental biologists and I strongly support publication.

We thank the reviewer for their positive comments.

Minor issues

Abstract. I found the description that cells orient “posterior migratory forces toward the cluster interior” confusing, as the direction of migration is radially away from the cluster center.

We have reworded to abstract to reflect this-

(Page 1 lines 14-16)-Here, using live imaging of the developmental migration program of *Drosophila* primordial germ cells (PGCs), we show that cluster dispersal is accomplished by stabilizing and orienting migratory forces.

Fig. 1A,H. Add information about when the endoderm EMT occurred.

We have added text indicating that the endoderm EMT occurred at $t = 14$ min in both Fig. 1A and 1B. Both timelapse experiments are shown at equivalent developmental time periods, using endoderm EMT as a reference point. We regret that it was unclear that all migration data were quantified post endoderm EMT. As such, we have noted in the Y axis of Fig. 1H that relative distance to endoderm center is taken post endoderm EMT. We have also added that migration speed, straightness, and distance are quantified post endoderm EMT.

I'd love to see Fig S1/S2 included in the main Figures—this is among the most impressive uses of the Rho sensor I have ever seen. Is Rho itself polarized?

We have combined Fig. S1/S2 and moved them into the main text as Fig. 2. We had previously reported that RhoA protein was polarized in WT and uniform in *tre1* PGC clusters in fixed embryos (1). The localization of Anillin-RBD has lead us to the new conclusion that Tre1 regulates the orientation and stability of an intrinsic RhoA polarity.

This is now included in the text as follows

(page 6, Line 149-151)- Thus, as opposed to our previous interpretation that Tre1 generates RhoA polarity^{12,22}, Tre1 regulates the orientation of an intrinsic RhoA polarity.

Figure 3. Do individual transplanted wildtype cells prematurely penetrate the endoderm?

We did not observe any premature entry into the endoderm. We have previously shown that PGC entry migration through the endoderm requires the cell contact loosening that occurs at EMT (2). Further, there is no evidence that *tud* mutant embryos prematurely enter EMT.

We have clarified this in the text

(page 8, line 228-231)-Strikingly however, transplanted WT PGCs, now unconstrained by other PGCs within the endoderm cavity, directionally migrated toward the periphery of the pre-EMT endoderm without prematurely crossing (Fig. 4b,d-f, Movie S9, S10), suggesting that WT PGCs utilize migratory forces to separate rather than a distinct contractile program.

p. 10, line 273. I might suggest slightly toning down the “sufficient” conclusion. They can clearly reverse migration with Rho activation in a somewhat artificial way but saying this demonstrates sufficiency is a bit much.

We reworded the text as follows-

(page 11, line 283-284)-Our results suggest that local RhoA activation and likely subsequent myosin II recruitment can specify the direction of PGC migration

We have also changed the figure title to reflect this-
Myosin II is necessary for PGC dispersal and its local accumulation can redirect migration.

Reviewer #2

New imaging methods and analysis give an exquisitely detailed description of the detachment and migration of PGCs away from their cluster. However, most of the observations are not entirely novel and despite a clever discussion, the main message of the article seems redundant to previous work which strongly reduced my enthusiasm.

We thank the reviewer for the positive comments on our analysis and hope that the clarifications and experiments we have performed clearly demarcate the advances presented here over our previous work. Specifically, our *in vivo* observations with polarity reporters allowed us to revise our previous model for the effect of *tre1* on germ cell polarity. Opposed to our previous interpretation that Tre1 generates RhoA polarity, we demonstrate now that Tre1 regulates the orientation of an intrinsic RhoA polarity.

Major concerns:

*- The first major conclusion of the manuscript is that forces formed at the back of PGCs direct their migration out of the cluster. This is inferred from the localization of contractility in the actomyosin cytoskeleton (observed by different markers: pMLC2 and tagged Myosin II) and by PGCs morphological changes. There is no direct measurement of forces, which is obviously hard if not entirely impossible in this *in vivo* system. Still, the description should be more precise, I understand that it is extremely likely that contractility and morphological changes are reporter of forces, but it is not clear in which direction the forces is generated although the observed migration correlates with the generation of propulsive forces at the back.*

Unfortunately the depth of the PGCs at this stage in development, as the reviewer has alluded to, make direct force measurements with laser ablation technically challenging. Thus our work assumes that migrating PGCs exert pulling forces on cell-cell adhesions at the rear as the PGC migrates. However, in support of our assumption, 3D models of migration, in which cells exhibit stable myosin II at the rear and do not possess notable protrusions (akin to PGCs in *Drosophila*), have noted the presence of posterior pulling forces (3,4).

We now clearly state that this is a caveat in the discussion as follows

(lines 372-376, page 14) A caveat to our model is that we have not directly shown that migrating PGCs exert posterior pulling forces, as this is technically challenging at the depth where PGC cluster dispersal occurs. However, posterior pulling forces have been clearly demonstrated in various cell types utilizing a rearward driven 3D migration mode which closely resembles PGC migration in *Drosophila*^{44,45}.

Optogenetics experiments show that Rho activation can modify the directionality of PGCs' migration. That said, the optogenetic results are not very compelling. Indeed, the effect is weak as less than a third of the PGCs change directionality when Rho is locally activated. Indeed 72%, i.e. a vast majority did not. The author should at least explain why the majority does not change direction. An important control, would also be to show that local activation of Rac

(photoactivable Rac is available in Drosophila) is not able to change directionality (as the migration of PGCs was shown to be independent of Rac activation by the Lehmann group (Kunwar et al., PLoS Biology 2003)). (Also: there is no statistical analysis on Fig.4g).

We agree that our reversal efficiency is poor in relation to prior uses of the Optogenetic RhoA activation system in epithelia (5) and an explanation is warranted in the text. As opposed to prior uses of optogenetic RhoA activation which manipulate signaling on the most superficial layer of the organism, our experiments are carried out at a variable depth of 50 – 80 μm when PGCs migrate in the mesoderm. Given the exponential dependency of power with depth, one reason for the low success rate is due to variations in depth leading to inadequate power delivery, as we maintained consistency by utilizing the same laser settings in all experiments. Recent work has also shown that the Cry2-Cibn optogenetic systems (utilized by Optogenetic RhoA activation) are poorly actuated by 2P illumination (6). Membrane recruitment of cry2-mScarlet was about 3 fold lower when comparing 2P vs. 1P activation (Fig. 2 in reference), likely leading to the lower efficiency observed in our experiments at greater depth. This study also indicated that LOV based optogenetic systems, such as PA-Rac1, are essentially insensitive to 2P activation (Fig. 6 in reference). Experiments with PA-Rac1 are likely to require new constructs utilizing FRET assisted 2P actuation, as shown in (6), and are beyond the scope of this study.

We have added statistics to Fig. 5g (Fig. 4g is now Fig. 5g), using a Fisher's exact test.

We have added the following statements to the text-

(pages 10-11, lines 281-283) The relatively low reversal rates we observed in our experiments are likely due to the depth of PGCs at this developmental stage (~50-80 μm) and the decreased 2P actuation efficiency of CRY2-CIBN systems³⁷.

- *The second major conclusion is that Tre1 is mediating this contractility. This is not really new as previous work from the same group showed that Tre1 acts through Rho1 (Kunwar et al., PLoS Biology 2003) and that Tre1 polarizes Rho1 activity (Kunwar et al., JCB 2008; Leblanc and Lehmann, JCB 2017)). Since Myosin II is directly activated downstream of Rho1, the result is not surprising although it is obviously important to demonstrate that the canonical Rho1 pathway is involved.*

Our prior work indeed suggested that Rho1 works in the same pathway as Tre1 and that Tre1 is involved in polarizing Rho1 but did not uncover where Rho1 was active. However, in this current work we have now revealed that PGCs have the intrinsic ability to spontaneously polarize Rho1 activity independently of Tre1. Thus in contrast to previous conclusions using fixed material, Tre1 does not have an essential role in generating polarity. Rather, our new results demonstrate that Tre1 stabilizes and orients existing polarities. We also now show that PGC-PGC separation is driven by orienting migration away from cluster rather than by a distinct set of Rho1 signaling events used solely for removing cell contacts, as has been shown in other cell-cell separation events from epithelia.

We have added the following clarifying statement to the text

(Page 6, lines 149-151)-Thus, as opposed to our previous interpretation that Tre1 generates RhoA polarity^{12,22}, Tre1 regulates the orientation of an intrinsic RhoA polarity.

- Again, the conclusions about adhesion are not entirely novel as the Lehmann group already showed that the modulation of DE-cadherin was not impacting on migration (Kunwar et al., JCB 2008). The reasons to test neuroglian are not very compelling.

The reviewer is correct in that we had previously shown that decreasing DE-cadherin levels were sufficient to disperse *tre1*^{-/-} PGC clusters but it has remained unclear if DE-cadherin levels are modulated in WT PGC cluster dispersal. This is important because modulation of adhesion has been debated as a mechanism of cell dissemination in the epithelial separation field and it is unclear if cell cluster dispersal requires such modulation. Here, using live imaging, we now reveal that DE-cadherin levels remain constant on the membrane during separation, which is in agreement with work indicating the maintenance of E-cadherin during malignant cell separation (7). This observation has also been highlighted as significant by the other reviewers.

We have added the following clarifying statement in the text (page 12, lines 314-316)- Thus, although a decrease in E-cadherin levels is sufficient to disband *tre1* PGC clusters¹², overt E-cadherin regulation appears to be dispensable for WT cluster dispersal.

We reasoned that investigating the effect of increased adhesion levels was similarly important, as this has been shown to block cell-cell separation during other developmental EMT processes. We utilized Neuroglian because it is sufficient to ectopically adhere normally non-adherent insect S2 cells and associates with a different set of cytoplasmic effectors than E-cadherin. The Neuroglian overexpression experiments were sufficient to increase PGC-PGC adhesion yet PGCs were still able to disperse, thus allowing us to conclude that PGCs can overcome increased cell-cell adhesion in general rather than limiting our conclusions to DE-cadherin or DE-cadherin cytoplasmic effectors.

We have added additional motivation in the text as follows (Page 12, lines 320-324)- We chose these molecules because they represent calcium dependent (E-cadherin) and independent (Neuroglian) means to increase adhesion, are sufficient to ectopically adhere insect S2 cells, and associate with different cytoplasmic effectors^{42,43}, thus informing us more generally how altering adhesion affects PGC dispersal.

*- The data about the requirement for a collective movement of PGCs in opposite direction is interesting, but I am wondering if there can be other explanations (for example that when WT PGCs adhere to multiple *Tre1*^{-/-}, they cannot migrate properly as they have a charge to move). I am not entirely convinced that it is the collectiveness of movement that is important and I don't know how to test this in a more compelling way.*

We agree that WT PGCs may adhere poorly to *tre1*^{-/-} PGCs, leading to defects in transmigration. However, we have measured qualitatively similar levels of E-cadherin, the chief cell-cell adhesion molecule in PGCs, in WT and *tre1*^{-/-} PGCs, suggesting that this may not be the case. We have also previously shown that defects in PGC adhesion typically manifest in PGCs being left outside of the endoderm on the surface of the embryo (8). We did not observe such defects when transplanting WT PGCs in *tre1*^{-/-} embryos.

We have added the following qualifying statement to the text (page 13, lines 348-352)-Defects in WT PGC transmigration from the interior of *tre1* clusters could alternatively result from aberrant adhesion between WT and *tre1* PGCs, leading to inefficient motility. However, we did not find significant differences in the chief PGC-PGC adhesive molecule, E-cadherin, between WT and *tre1* PGCs during live imaging of endogenously tagged E-cadherin (Fig. 6c).

- The use of the word "sensor" for the Rho, Rac or Cdc42 activity when using a simple binding domain is, in my opinion, improper. Indeed, FRET sensors working in vivo in Drosophila for Rac and Cdc42 exist (I don't think that there is a Rho sensor). These are indeed activity sensor as they rely on the proportion of active GTPase, although they also have some caveats. Binding domains are recruited by active Rho GTPase, but they display the localization of active GTPase properly only if the proportion of binding domain is in the range of the number of active Rho GTPase. If the binding domain is overabundant, it will be localized at regions that are not associated with active GTPase, and can even bind possibly to other GTPases. I know that most of the probes used here were well characterized by the group of Susan Parkhurst but their efficiency might dramatically differ from one cell type to another. As such, I think that the data of the anillin binding domain are convincing but the Rac and Cdc42 binding domains seem largely overexpressed which may impair the observation of the site of their active target GTPase.

- Also, again, previous work from the Lehmann group already showed that Rac and Cdc42 are dispensable for PGC migration (Kunwar et al., PLoS Biol 2003).

We agree and have modified the phrase RhoA sensor to Anillin RhoA-GTP binding domain (RBD) throughout the text and figures. We have also used Cdc42-GTP and Rac-GTP binding domains in place of biosensors in the text and figure S3. Lastly, we added the following qualifying statement-

(page 6, lines 161-163) GFP tagged Cdc42-GTP and Rac-GTP binding domains²⁹ remained cytoplasmic and were uniformly distributed in WT clusters, suggesting that protein activity was unpolarized (Fig. S3). However, we are unable to rule out subtle differences in localization due to saturation.

We have also further characterized the Anillin RBD in PGC clusters by overexpressing dominant negative RhoA, WT RhoA, and constitutively active RhoA (Fig. S2). Dominant negative RhoA decreases Anillin RBD enrichment in the cluster center relative to WT RhoA (Fig. S2f), while constitutively active RhoA did not abolish polarity.

Reviewer #3

This is a novel study using a number of cutting-edge tools to provide insights into the mechanisms of collective cell dispersal during development. There are implications for collective metastasis in cancer, as well as other migratory collectives during development. This manuscript

will be of wide interest. The experiments appear to be rigorous, well-controlled, and are overall convincing. I have a few specific comments that would help clarify the results and discussion.

We thank the reviewer for their positive comments.

Comments:

1. *This statement (lines 107-108) was confusing: “Subsequently, the foci were either tugged off of other PGCs and incorporated into trailing tails or were snapped off and left behind in the endoderm cavity (Fig. 1a).” I am unsure what the authors are referring to and this is not clear in the figure or in the movies. Additional labels on the figure, or panels showing this more clearly, may be needed.*

We apologize and agree that the cell behavior we describe is not immediately clear in the figure as the 3D movement of individual PGCs is difficult to capture in a single Z slice. Due to space constraints on the figure, we have now highlighted tail retention (Fig. S1a) or detachment (Fig. S1b) from PGCs in different Z slices from the same cluster shown in Fig. 1a. This is now included as Movie S3.

2. *The RhoA biosensor was validated in S2 cells, but not in PGCs (Fig. S1). It would be more convincing if the authors had co-expressed RhoN19 (DN) and RhoV12 (CA) with the biosensor in PGCs (if possible), to show similar relocation of the sensor to the cytoplasm with DN Rho and to the membrane with CA Rho. It is reassuring that two effectors of Rho (Dia and Rock) show similar localization with the RhoA biosensor (Fig. S2).*

We have performed additional experiments (Fig. S2) to characterize the RhoA biosensor, now referred to as Anillin-RBD, in PGCs by co-expressing UAS driven RhoA WT, RhoA G14V (CA), and RhoA T19N (DN) with a germ cell specific Nos-Gal4VP16. We did not observe complete dampening of RhoA activity with RhoA-DN, likely due to variable expression levels. However, we observed a decreased enrichment in the center of PGC clusters expressing RhoA DN as compared to RhoA WT or RhoA-CA (Fig. S2f), reflecting a suppression of RhoA polarity. Interestingly, RhoA-CA overexpression did not enhance or disrupt polarity relative to RhoA-WT (Fig. S2f), suggesting PGCs may be capable of self-organizing polarity, akin to yeast polarizing in the context of Cdc42-CA expression (9).

These experiments have been added to the main text as follows

(page 6, lines 141-145)-This enrichment overlapped with PGC posterior membranes (Fig. 2a), supporting our previous observations of posterior contraction (Fig 1a,c), and was reduced when overexpressing a dominant negative RhoA relative to WT RhoA (Fig. S2c,e,f). Overexpression of constitutively active RhoA did not perturb this distribution (Fig. S2d,f), suggesting an ability to self-organize polarity.

3. *In Figure 2e-h, the authors show that myosin-II-GFP is polarized in the PGC cluster in WT but not *tre1*^{-/-} mutants. This is convincing, and is followed up in Figure 3 with the transplantation experiments. However, it seems that myosin-II-GFP is highly dynamic in *tre1*^{-/-} mutant embryos (Fig. 2f and Movie S6), suggesting that the contractility is unstable and thus PGCs cannot contract and disperse. I am unsure that Fig. 2h really captures the quantification of this dynamic*

*myosin-II-enrichment; could the authors perform kymographs on their existing movies, across the PGC cluster in *tre1*^{-/-} mutant embryos to show fluctuations in GFP enrichment over time? Or is this just not possible because of the 3D nature of 30 cells in the embryo?*

We thank the reviewer for pointing this out. These dynamics are indeed difficult to capture for two reasons (1) the data are smoothed out when averaging over many clusters and (2) the dynamic range of the myosin II intensity levels in *Tre1* clusters is lower in relation to WT (Both are normalized in the same manner to background). We have added individual Myosin II heatmaps for the individual WT and *tre1*^{-/-} PGC clusters shown in Fig. 3e,f in Fig. S4a,b where these dynamics are more readily appreciated. We have scaled the heatmaps in Fig. S4a,b differently to illustrate these dynamics.

4. In the last section of the Results (lines 334-337; Fig. 6c-e), the wording is a bit confusing and could be clarified.

We have clarified the text as follows

(page 13, lines 352-355) Groups of WT PGCs (≥ 3 WT PGCs) had reduced contact with *tre1* PGCs and migrated outwards concurrently (Fig. 7c), detaching from *tre1* PGC clusters with a ~ 1.4 fold reduction in frequency compared to controls (Fig. 7c-e, Movie S21). These results suggest that increasing cell-cell coordination improves cluster dispersal efficiency.

5. The model in Fig. 7 only focuses on individual WT or *tre1*^{-/-} mutant PGCs, and lacks a figure legend. If the point is for group migration and contraction to help disperse the cluster of PGCs, along with individual PGC polarity, it would be helpful to see that in the summary panels. Indeed, the influence of collective behavior is not highlighted in this summary. We have now illustrated the motility of multiple PGCs in WT and *tre*^{-/-} backgrounds in the summary figure (Fig. 8) and have also added a legend to describe how coordinated outward movements aid dispersal.

Minor comments:

1. In Fig. 1a-b, there are asterisks on a few cells. I assume these are the cells shown in the movies, but this could be mentioned in the figure legend.

We have updated the legend to include that the cells marked with asterisks are shown in Movie S2 and S4.

2. As far as I can tell, Movie S7 is missing a call-out in the text. I assume this movie should be mentioned around lines 200-202.

We have included a callout for this Movie (now Movie S8).

3. The statistics for Fig. 4g (*OptoRhoGEF* migration reversal) are missing on the panel.

We have performed the statistical comparison using a Fisher's exact test and have added an asterisk noting that this is significant ($P < .05$).

4. Fig. 6e, the y-axis is labeled "% Successful transmigration" but the numbers are from 0-1.0 and not 0-100%. I assume this is mislabeled.

We have corrected the Y-axis in this figure.

- 1) Kunwar, P.S. *et al.* Tre1 GPCR initiates germ cell transepithelial migration by regulating *Drosophila melanogaster* E-cadherin. *The Journal of Cell Biology* **183**, 157 (2008).
- 2) Seifert, J.R.K. & Lehmann, R. *Drosophila* primordial germ cell migration requires epithelial remodeling of the endoderm. **139**, 2101-2106 (2012).
- 3) Poincloux, R. *et al.* Contractility of the cell rear drives invasion of breast tumor cells in 3D Matrigel. **108**, 1943-1948 (2011).
- 4) Shih W, Yamada S. Myosin IIA dependent retrograde flow drives 3D cell migration. *Biophys J.* 2010;98(8):L29–L31. doi:10.1016/j.bpj.2010.02.028
- 5) Izquierdo, E., Quinkler, T. & De Renzis, S. Guided morphogenesis through optogenetic activation of Rho signalling during early *Drosophila* embryogenesis. *Nature Communications* **9**, 2366 (2018).
- 6) Kinjo, T., Terai, K., Horita, S. *et al.* FRET-assisted photoactivation of flavoproteins for in vivo two-photon optogenetics. *Nat Methods* **16**, 1029–1036 (2019).
- 7) Eliah R. Shamir, Elisa Pappalardo, Danielle M. Jorgens, Kester Coutinho, Wen-Ting Tsai, Khaled Aziz, Manfred Auer, Phuoc T. Tran, Joel S. Bader, Andrew J. Ewald; Twist1-induced dissemination preserves epithelial identity and requires E-cadherin. *J Cell Biol* 3 March 2014; 204 (5): 839–856.
- 8) DeGennaro, M. *et al.* Peroxiredoxin Stabilization of DE-Cadherin Promotes Primordial Germ Cell Adhesion. *Developmental Cell* **20**, 233-243 (2011).
- 9) Wedlich-Soldner, Roland, *et al.* "Spontaneous cell polarization through actomyosin-based delivery of the Cdc42 GTPase." *Science* 299.5610 (2003): 1231-1235.

REVIEWERS' COMMENTS

Reviewer #2 (Remarks to the Author):

The response to my comments and the changes applied to the manuscript answer to most of my issues. I liked the care with which the authors clarified several aspects and the modifications that they applied to the text. In particular, I found really useful the explanations regarding the technical limitation of optogenetic experiments or regarding some aspects of the Tre or neurglian experiments that I overlooked. The responses to the other reviewers' comments also seem adequate.

Overall, I really like the manuscript as it is now, however I still have some concerns regarding the novelty of the findings. I thank the authors to have made the efforts to outline the differences with their previous work in their rebuttal. As mentioned above, some clarifications helped me to better understand the novelty of their finding (in particular regarding Tre). Though, I still feel that this work is a refinement of previous work.

As I write in my comments to the editor, I would have liked to have a forum to discuss this matter with my fellow reviewers. I understand that I may be overly critical, but they also may have overlooked this issue. A direct discussion would have possibly solved that. Unfortunately, we don't have access to such a forum. Hence, if my colleague reviewers and the editor are not concerned by the lack of novelty, I would definitively support the publication of this manuscript as the data are of high quality and clearly backup the message of this study.

Reviewer #3 (Remarks to the Author):

In this manuscript, Lin and colleagues investigate how primordial germ cells (PGCs) are collectively dispersed during development. The authors find that polarized positioning of contractile actomyosin forces facilitates coordinated symmetric pulling for cluster dispersal. In this revised manuscript, the authors performed new experiments, added clarifying text, and clarified their model. These changes have strengthened the manuscript and made it much clearer. The authors have satisfactorily addressed all of my concerns and those of the other reviewers.